# Adaptive-Solver Framework for Dynamic Strategy Selection in Large Language Model Reasoning

## Abstract

As the field of artificial intelligence evolves, Large Language Models (LLMs) are showcasing impressive ability in handling complex reasoning tasks. Researchers have developed various techniques utilizing LLMs to tackle these challenges. In real-world situations, problems often span a spectrum of complexities. Humans inherently adjust their problem-solving approaches based on task complexity. However, most methodologies that leverage LLMs tend to adopt a uniform approach: utilizing consistent models, prompting methods, and degrees of problem decomposition, regardless of the problem complexity. Inflexibility of these methods can bring unnecessary computational overhead or sub-optimal performance. To address this problem, we introduce an Adaptive-Solver framework. It strategically modulates solving strategies based on the difficulties of the problems. Given an initial solution, the framework functions with two primary modules. The initial *evaluation* module assesses the adequacy of the current solution. If improvements are needed, the subsequent *adaptation* module comes into play. Within this module, three key adaptation strategies are employed: (1) *Model Adaptation*: Switching to a stronger LLM when a weaker variant is inadequate. (2) *Prompting Method Adaptation*: Alternating between different prompting techniques to suit the problem's nuances. (3) *Decomposition Granularity Adaptation*: Breaking down a complex problem into more fine-grained sub-questions to enhance solvability. Through such dynamic adaptations, our framework not only enhances computational efficiency but also elevates the overall performance. This dual-benefit ensures both the efficiency of the system for simpler tasks and the precision required for more complex questions. Experimental results from complex reasoning benchmarks reveal that the *prompting method adaptation* and *decomposition granularity adaptation* within the Adaptive-Solver framework enhance performance across all tasks. Furthermore, the *model adaptation* approach significantly reduces API costs (up to 50%) while maintaining superior performance.[1]

## 1 Introduction

Large Language Models (LLMs) exhibit remarkable proficiency across a myriad of reasoning tasks. However, while the potential of LLMs in addressing intricate problems is undeniable, the quest to identify the most effective problem-solving strategy to maximize their performance remains largely untapped. In our pursuit to address this problem, we turn to draw inspiration from the innate problem-solving approaches employed by humans. The human cognitive framework consists of two distinct systems: *System 1* for intuitive thinking, and *System 2* for deeper, analytical reasoning (Sloman, 1996; Daniel, 2017). These systems are utilized dynamically and adaptably, catering to a range of problem complexities, thereby ensuring both efficiency and accuracy in problem-solving.

Likewise, when faced with complex challenges, humans often break down the problem into more detailed sub-questions, ensuring a lucid formulation of the task. For simpler question, a direct, singular line of reasoning is typically employed. If their initial solution not meet expectations, humans naturally pivot their approach in pursuit of a more effective resolution. Recognizing the multifaceted

---

[1]We will release all our code upon acceptance to facilitate research on this line.

nature of real-world challenges and drawing inspiration from human problem-solving methodologies, it stands to reason that machines too should be equipped with the capacity to dynamically adjust their problem-solving strategies. This adaptation might encompass variations in the underlying LLM models, granularity in problem-decomposition techniques, or alternative prompting methods.

Current research trends often employ a static solver[2], neglecting the distinct characteristics of individual problems. This inflexibility in adjusting the solver to diverse problems can result in unnecessary resource consumption and sub-optimal performance. For example, GPT-4, while possessing remarkable capabilities, comes with a significant API cost. Utilizing a more cost-effective model for simpler queries can be a strategy to reduce expenses. Additionally, at the problem-solving method layer, the Chain-of-thought (CoT) (Wei et al., 2022) prompts LLMs to generate an intermediary reasoning process to yield reliable results. However, its reliance on a single-turn of reasoning, without explicit sub-problem decomposition, makes it less suitable for complex challenges. To enhance CoT, Self-consistency (SC) (Wang et al., 2023c) generates results multiple times and selects the answer through majority voting, while Least-to-most (L2M) decomposes the main problem into distinct sub-problems. Despite improved performance, these methods face limitations: they either use a fixed sampling quantity or lack the flexibility to adjust the granularity of problem decomposition—such as modifying the number of sub-problems—based on the problem's complexity. A decomposition that is too coarse may oversimplify the main question, while an excessively detailed breakdown can increase the risk of decomposition errors. Balancing granularity is essential to optimize problem-solving effectiveness. Thus, we argue that distinct problems necessitate dynamically customized solvers to achieve both optimal cost-efficiency and enhanced performance.

In response to the clear demand for dynamic problem-solving methods, we propose the Adaptive-Solver (AS) framework. This innovative approach customizes the problem-solving strategies of LLMs, adapting based on the task's complexity and an initial evaluation of the solution. The AS framework is structured around two core modules: the *evaluation* module and the *adaptation* module. The *evaluation* module assesses the current solution's efficacy, determining whether it meets the problem-solving standards. Should the solution not meet the requisite quality, the *adaptation* module is triggered, adjusting the solving strategy for the following phase. Our Adaptive-Solver framework adopts three pivotal adaptation strategies : (1) *Model Adaptation*: Shifting to a more powerful, albeit resource-intensive, LLM when necessary; (2) *Prompting Method Adaptation*: Varying the prompting techniques to better align with the complexity of the problem; (3) *Decomposition Granularity Adaptation*: Breaking down complex questions into finer sub-questions to enhance problem formulation. A characteristic of our framework is its adaptability: both modules are architecturally independent, enabling a broad spectrum of potential implementations, thereby underscoring its applicability across diverse scenarios. We further explore the potential of combined strategies by introducing a hybrid implementation that integrates *prompting method adaptation* with *decomposition granularity adaptation*. This innovative framework propels us into a promising direction in dynamic strategy selection for LLMs. From a broader perspective, every solver – whether it's model, prompting, decomposition, or augmented tools – can be considered a potential component in the pool. LLMs, equipped with the AS framework, demonstrate the capability to dynamically combine selected components, forging pathways to optimal solution paths.

Extensive experiments across 8 reasoning tasks corroborate the effectiveness of the Adaptive-Solver and draw several crucial findings: (1) The Adaptive-Solver consistently elevates performance across every task. This underscores the merit of dynamic strategy selection in enabling LLMs to select the optimal reasoning technique for multifaceted challenges. (2) Leveraging the *model adaptation* strategy notably reduce API cost (up to 50%), while upholding a superior performance.

Our contributions can be distilled into the following key points: (1) We introduce the Adaptive-Solver framework. It is adept at strategically selecting the optimal solving methodologies tailored to the intrinsic characteristics of a given problem. (2) We propose three versatile adaptation strategies concerning model selection, prompting methods, and decomposition granularity. Furthermore, we explore the synergistic effects of their combined application. (3) Experiments underscore the superiority of the Adaptive-Solver framework, demonstrating marked enhancements in computational efficiency and performance outcomes.

---

[2]In this context, a solver encompasses all elements integral to problem-solving, including the LLM model, prompting techniques, decomposition strategies, and so forth.

## 2 RELATED WORK

**Reasoning with LLM prompting.** It is widely recognized that complex reasoning problems are quite challenging for language models. Such problems include mathematical reasoning (Lu et al., 2023; Cobbe et al., 2021b), commonsense reasoning (Talmor et al., 2018), symbolic reasoning (Wei et al., 2022) and logical reasoning (Creswell et al., 2023). The recently proposed CoT (Wei et al., 2022) prompting significantly enhances the complex reasoning capabilities of LLMs, by generating intermediate reasoning steps to obtain the answer. Similarly, (Kojima et al., 2022) proposes Zero-CoT to elicit reasoning step generation without exemplars. PAL (Gao et al., 2023) and PoT (Chen et al., 2022) generate programs to represent the reasoning process and utilize a code interpreter to execute the programs. CoT has inspired diverse prompting methods aimed at further enhancing the complex reasoning capabilities of LLMs. Among these works, there are two prevailing technical approaches. The first type of methods adopt the idea of "divide and conquer". PS prompting (Wang et al., 2023a) devises a plan to divide the entire task into smaller subtasks, and then carry out the subtasks according to the plan. Besides, some methods (Zhou et al., 2023; Khot et al., 2023) decompose the main problem into simpler sub-problems to solve. The second type of methods adopt the idea of "try more". SC (Wang et al., 2023c) decoding strategy improves CoT by sampling multiple solutions in a single round and determining the final answer through majority voting. PHP (Zheng et al., 2023) solves problems iteratively over multiple rounds and utilizes generated answers as hint to guide the subsequent responses. Besides, some works (Yao et al., 2023; Xie et al., 2023) sample multiple responses for each step and integrate step-wise self-evaluation to guide the generation of a whole solution. However, most of the existing works construct a fixed solver for different problems, regardless of their varied complexity, which may result in unnecessary computational overhead or sub-optimal performance. We introduce Adaptive-Solver framework that can adapt the solver to different problems. Note that, PHP adapts the number of solving rounds based on the consistency of recent consecutive answers, yet using the same LLM model and prompting method. We aim to propose a general framework the supports adapting a solver from various aspects.

**Automated feedback for LLMs.** Another relevant research area is providing automated feedback to the LLM's response. (Pan et al., 2023) divide automated feedback into two types according to the sources: self-feedback and external feedback. Self-feedback denotes the feedback originated from the LLM itself, i.e, self-evaluation (Madaan et al., 2023; Weng et al., 2023; He et al., 2022). External feedback represents the feedback derived from external models (Wang et al., 2023b), tools (Gou et al., 2023), metrics (Jung et al., 2022) and knowledge bases (Yu et al., 2023). The *evaluation* module in our framework can be implemented based on various automated feedback methods. Since we focus on the *adaptation* module, for simplicity, we adopt a self-consistency-based metric (i.e., *consistency*) (Wang et al., 2023c) to evaluate the answer.

## 3 THE SOLVER-ADAPTIVE FRAMEWORK AND ITS INSTANTIATIONS

**Overview.** Our Adaptive-Solver (AS) framework integrates multiple solvers and dynamically adapts to the most suitable solving strategies according to the problem characteristic. This framework comprises two main modules: *evaluation* module and *adaptation* module. The framework's workflow is depicted in Figure 1(a): 1) Given a problem, the solutions are generated by the current solver. 2) The *evaluation* module assesses whether the solutions successfully meet the evaluation criteria. If the criteria are satisfied or the maximum predefined number of solving attempts is reached, the solving process terminates. 3) If the criteria are not met, the *adaptation* module will be activated to adjust the solver, and then the process proceeds to the next solving round. Within this module, three key adaptation strategies are designed. *Model Adaptation* (shown in Figure 1(b)): Switching to a more advanced LLM to ensure the accuracy of solving complex problems. *Prompting Method Adaptation* (shown in Figure 1(c)): Alternating between different prompting techniques to suit the problem's characteristic. *Decomposition Granularity Adaptation* (shown in Figure 1(d)): Breaking down a complex problem into more refined sub-questions to reduce the difficulty of each sub-question.

This new method is a notable advancement in dynamic strategy selection for Large Language Models. Viewing from a more expansive angle, every solver—be it model, prompting, decomposition, or augmented tools—may be regarded as a potential element in the component pool. LLMs, armed with the Adaptive-Solver (AS) framework, showcase the ability to dynamically compose selected components, thereby paving the way for optimal solution paths.

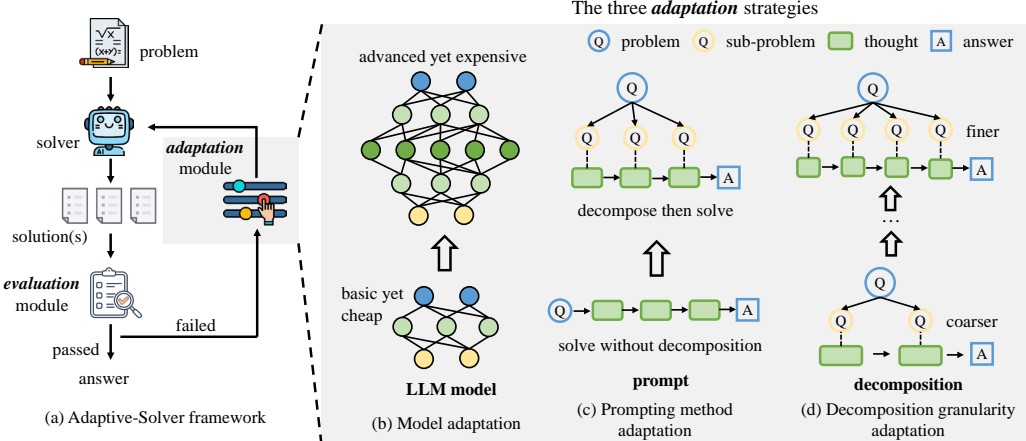

Figure 1: The Adaptive-Solver (AS) framework and the three adaptation strategies.

## 3.1 EVALUATION MODULE

The *evaluation* module aims to evaluate whether the current solver is sufficient to resolve the problem, and decide when to adapt the solver. Our framework exhibits strong flexibility, capable of supporting different types of implementations for the *evaluation* module, including internal evaluation and external evaluation. Internal evaluation involves the language model itself reflecting upon or assessing the solution. However, this method relies heavily on the self-evaluation ability of the LLM. External evaluation involves evaluating the solution or the answer using external models, tools, or metrics. This paper adopts a self-consistency-based metric to evaluate the answer. (Wang et al., 2023c) found that the *consistency* (in terms of % of decodes agreeing with the final aggregated answer) is highly correlated with accuracy. This enables us to leverage *consistency* to estimate the likelihood of the current answer being correct and reflect the confidence of model prediction. Therefore, in our implementation of the proposed framework, each solver samples $N$ diverse solutions during a single solving round and then the metric *consistency* is calculated. If the *consistency* (i.e., # of the most consistent answer / $N$) reaches a predefined threshold $\theta$, the solving process terminates. This paper, unless otherwise specified, sets $N$ to 3 and the threshold to 1.

## 3.2 ADAPTATION MODULE

The *adaptation* module aims to address the shortcomings of the "one solver for all problems" strategy. It dynamically adapt the solver to different problems. This enables it to identify an appropriate solver for each problem, one that helps reduce computational costs or enhance performance. We adopt a straightforward approach to implement solver adaptation, which involves designing a list of solvers and switching to the next solver in the list from the current one when adaptation is needed. Therefore, we denote an adaptation strategy as a list $\mathcal{A}$ of solvers:

$$\mathcal{A} = [\mathcal{S}_1, \mathcal{S}_2, ..., \mathcal{S}_n], \mathcal{S}_n = (m_n, p_n, d_n, ...)$$

where $\mathcal{S}_n$ represents the $n$-th solver, and each solver $\mathcal{S}_n$ can be represented as a tuple of elements such as LLM model denoted as $m_n$, prompting method denoted as $p_n$, and decomposition granularity denoted as $d_n$. For simplicity, we represent $\mathcal{S}_n$ with only the adjustable elements. For example, the *model adaptation* strategy is represented as $[m_1, m_2, ..., m_n]$.

Given the list $\mathcal{A}$, the current solver is adjusted by switching to the next solver until either all solvers are tried or the evaluation criteria are satisfied. If the criteria are met, the current solver is chosen. If none of the solvers meet the criteria, two strategies are available to select the final solver: 1) Choose the last solver in the list. 2) Select the solver with the highest *consistency*, as calculated in the *evaluation* module. In the case of multiple solvers having the same highest *consistency*, the most recently invoked solver is selected. We use the first strategy for the *model adaptation* and *prompting method adaptation*, and the second strategy for the *decomposition granularity adaptation*.

**Model adaptation**: Since many of the simple questions can be answered by sub-optimal LLMs, we endeavor to reduce the cost by only invoking stronger but more expensive LLMs for those complex problems, as illustrated in Figure 1(b). We explore two implementations of *model adaptation*, each with its corresponding solver list as: $\mathcal{A}_{M_1}$ = [GPT3.5*, GPT4][3] and $\mathcal{A}_{M_2}$ = [GLM2*, GPT3.5]. Among these, GLM2 (Du et al., 2022) is an open-source model that can be deployed locally. GPT3.5 (OpenAI, 2022) is a relatively cheaper closed-source model, while GPT4(OpenAI, 2023) is the highest-performing model but also comes with the highest usage costs.

**Prompting method adaptation**: Different prompting methods may be suitable for different problems. For example, (Zhou et al., 2023) found that, on the GSM8K dataset, for problems requiring a greater number of steps (i.e., 5 steps or more), L2M outperforms CoT significantly; while for problems requiring fewer steps, both perform similarly or even CoT performs better. Therefore, through *prompting method adaptation* (illustrated in Figure 1(c)), we hope that different problems can be matched to their own suitable prompting methods, so as to enhance the reasoning performance. We provide two implementations of *prompting method adaptation*, with the corresponding solver list as: $\mathcal{A}_{P_1}$ = [CoT*, L2M*] and $\mathcal{A}_{P_2}$ = [ZeroCoT*, PS*].

**Decomposition granularity adaptation**: To mitigate the constraint posed by L2M's inflexibility in adapting problem decomposition granularity, we introduce an adaptation method that tailors the decomposition granularity to each specific problem. Specifically, we design three different variants of L2M prompt, denoted as (L2M, $d_1$), (L2M, $d_2$) and (L2M, $d_3$), where $d_1$, $d_2$, $d_3$ represents the decomposition granularity, from coarser to finer. The only difference among them lies in the decomposition granularity (See appendix A.12.5 for specific prompts). As shown in Figure 1(d), as the solving process continues, the decomposition granularity in the prompt gradually becomes finer. The corresponding solver list is represented as: $\mathcal{A}_D$ = [(L2M*, $d_1$), (L2M*, $d_2$), (L2M*, $d_3$)].

Moreover, we explore one type of hybrid adaptation that combines *prompting method adaptation* and *decomposition granularity adaptation*. The corresponding solver list is represented as: $\mathcal{A}_{PD}$ = [CoT*, (L2M*, $d_1$), (L2M*, $d_2$), (L2M*, $d_3$)].

## 4 EXPERIMENTS

### 4.1 EXPERIMENT SETUP

**Datasets.** The proposed method is evaluated on 8 datasets from three categories of reasoning tasks. **Arithmetic Reasoning**: GSM8K (Cobbe et al., 2021a), SVAMP (Patel et al., 2021), AQuA (Ling et al., 2017), AddSub (Hosseini et al., 2014), SingleEq (Koncel-Kedziorski et al., 2015) and Multi-Arith (Roy & Roth, 2015); **Commonsense Reasoning**: CSQA (Talmor et al., 2019); **Symbolic Reasoning**: Last Letter Concatenation (LLC) (Wei et al., 2022) (See dataset details in Appendix A.1).

**Baselines.** 1) For *prompting method adaptation* and *decomposition granularity adaptation*, the baselines are various prompting methods. We include two types of prompting baselines: single-solution promptings solve problems in a single-turn, including ZeroCoT (Kojima et al., 2022), PS (Wang et al., 2023a), CoT (Wei et al., 2022) and L2M (Zhou et al., 2023); multi-solution promptings solve problems for multiple times, including CoT_SC (Wang et al., 2023c) and PHP (Zheng et al., 2023). We use GPT-3.5 as model for all these prompting methods.

2) For *model adaptation*, the baselines are the methods that using only weaker or stronger LLM, i.e., GPT 3.5 and GPT 4 in this paper. We uniformly employ ZeroCoT as the prompting method for all LLMs. (See implementation details in Appendix A.2)

### 4.2 MAIN RESULTS

**Results on Arithmetic Reasoning**. Table 1 reports the accuracy comparison of our method and existing single-solution and multi-solution methods on the arithmetic reasoning datasets. Our method has three variants: *prompting method adaptation* (denoted as AS-P), *decomposition granularity adaptation* (denoted as AS-D), and the combination of *prompting method adaptation* and *decomposition granularity adaptation* (denoted as AS-PD).

---

[3]In this paper, the method with * represent its self-consistency version, that is, the method samples multiple (specifically 3 in this work) solutions and determines the final solution by majority voting.

Table 1: Performance comparison on the arithmetic reasoning datasets. AS-P, AS-D and AS-PD respectively use the solver list $\mathcal{A}_{P_1}$, $\mathcal{A}_D$ and $\mathcal{A}_{PD}$ (§ 3.2). SS: Single-solution prompting, MS: Multi-solution prompting. The best results are boldfaced. We use GPT-3.5-turbo-0301 for all promptings.

| Type | Method | GSM8K | SVAMP | AQuA | AddSub | SingleEq | MultiArith | Average |
|------|--------|-------|-------|------|--------|----------|------------|---------|
| SS | ZeroCoT | 79.6 | 79.1 | 55.5 | 81.0 | 89.6 | 96.3 | 80.2 |
|    | PS | 78.8 | 80.0 | 59.8 | 87.3 | 93.9 | 96.0 | 82.6 |
|    | CoT | 80.8 | 80.6 | 57.5 | 88.9 | 96.1 | 98.3 | 83.7 |
|    | L2M | 77.7 | 83.1 | 52.4 | 90.5 | 93.3 | 93.3 | 81.7 |
| MS | CoT_SC | 84.3 | 82.2 | 63.4 | 90.6 | 96.3 | 97.8 | 85.8 |
|    | PHP | 85.5 | 81.4 | 63.6 | 86.4 | 92.8 | 98.2 | 84.7 |
|    | AS-P | 85.7 | 87.0 | 65.0 | 91.5 | 95.3 | **98.7** | 87.2 |
|    | AS-D | 87.5 | **89.0** | 63.3 | **92.9** | 95.6 | 98.4 | 87.8 |
|    | AS-PD | **88.6** | 88.5 | **66.5** | 92.2 | **96.9** | **98.7** | **88.6** |

**(1) Adaptive-Solver framework effectively enhances LLM's performance.** The three variants of our framework consistently outperform the baselines, with each dataset yielding its best result through one of our methods. In terms of average accuracy, AS-PD leads the best result of the baselines (i.e., 85.8%) by 2.8%. This result underscores the effectiveness of our approach in enhancing the reasoning capabilities of LLMs by dynamically selecting the most suitable prompting method and decomposition granularity. Furthermore, AS-PD, which combines both the *prompting method adaptation* and *decomposition granularity adaptation*, outperforms each individual strategy (AS-P and AS-D) in overall. This finding suggests that the synergistic use of various adaptation strategies can effectively complement one another, leading to an overall improvement in performance.

**(2) Multi-solution methods outperform single-solution approaches in overall performance.** Specifically, CoT_SC is designed to produce multiple solutions within a single solving round, while PHP adopts a multi-round solving approach but yields a single solution in each round. Our proposed methods integrate both the paradigms, enabling iterative problem-solving across multiple rounds, with generating multiple solutions at each iteration. All of these methods exhibit enhancements over single-solution prompting techniques, with our approaches delivering the most notable performance gains. These findings underscore the effectiveness of multi-solution promptings in significantly improving the model's accuracy.

**(3) Different prompting has its own strengths and weaknesses.** ZeroCoT and CoT represent methods that solve problems in a single stage without prior explicit planning or problem decomposition. PS and L2M represent methods that address problems in a two-stage approach, commencing with explicit planning or problem decomposition. ZeroCoT outperforms PS on two datasets while underperforming on the remaining four datasets. Similarly, CoT outperforms L2M on four datasets but doesn't do as well on the other two. This indicates that each of them has its own strengths and weaknesses, making them suitable for different types of problems.

**Results on Commonsense and Symbolic Reasoning**. Table 7 (Appendix A.3) reports the results on the commonsense reasoning dataset CSQA and the symbolic reasoning dataset LLC. Due to commonsense reasoning problems typically do not entail multi-step solving or problem decomposition, it becomes unnatural to apply L2M in this context. Therefore, we use two alternative zero-shot methods (i.e., ZeroCoT and PS) to implement *prompting method adaptation*, with the solver list as $\mathcal{A}_{P_2}$ (§ 3.2). From Table 7, we observe that the adaptive prompting method consistently outperforms not only zero-shot methods (i.e., ZeroCoT and PS) but also 4-shot methods (i.e., CoT and COT_SC).

## 4.3 MODEL ADAPTATION

The primary goal of *model adaptation* is to cut down on expensive API calls or computational resources required to solve a problem, while maintaining performance. We validate the effectiveness of *model adaptation* by examining both the performance and cost. We randomly select 200 samples from each dataset for the experiments of *model adaptation*. Additionally, the solver running on GPT-4 only generates one solution. Table 2 presents the results of *model adaptation* implemented with the model list [GPT3.5*, GPT4], while we also provide the implementation with the model list [GLM2*, GPT3.5], please see the results in Table 12 (Appendix A.8). ZeroCoT is the only prompting method used in the experiments of *model adaptation*.

Table 2: Model adaptation implemented with the model list [GPT3.5*, GPT4].

| LLM Model | GSM8K-200 | | SVAMP-200 | | CSQA-200 | | LLC-200 | |
|---|---|---|---|---|---|---|---|---|
| | ACC | Cost ($) | ACC | Cost ($) | ACC | Cost ($) | ACC | Cost ($) |
| GPT3.5* | 85.0 | 0.484 | 82.5 | 0.334 | 74.5 | 0.271 | 76.5 | 0.124 |
| [GPT3.5*,GPT3.5*] | 87.5 | 0.629 | 84.0 | 0.423 | 75.5 | 0.364 | 80.5 | 0.190 |
| GPT4 | 93.0 | 4.195 | 88.0 | 2.721 | 85.5 | 2.064 | 91.5 | 1.610 |
| [GPT3.5*,GPT4] | 93.5 | 2.167 | 90.0 | 1.261 | 82.0 | 1.036 | 91.0 | 1.034 |
| Acc. Gain / Saved $ | +0.5 / 48.3% | | +2.0 / 53.6% | | -3.5 / 49.8% | | -0.5 / 35.8% | |

**Model adaptation reduces the overall API cost while maintaining superior performance.** From Table 2, we can observe that: 1) GPT4 surpasses GPT3.5* by a significant margin, approximately 5-15%. However, this performance improvement is accompanied by a relatively higher cost, roughly 7-13 times more expensive. 2) [GPT3.5*,GPT4] outperforms [GPT3.5*,GPT3.5*] significantly, indicating that integrating only weaker model itself still has limitations in terms of performance. 3) [GPT3.5*,GPT4] performs at a comparable level to GPT4, and in certain cases, it even outperforms it slightly. Moreover, this combination significantly reduces the overall API cost, saving approximately 35-50% of API cost compared to the case using GPT4 alone.

**There exists a trade-off between the cost and performance in *model adaptation*.** We are interested at seeing how the cost and performance change when we modify the criteria of the *evaluation* module, achieved by adjusting both the sample size $N$ and the threshold $\theta$. The visualization in Figure 2 illustrates the relationship between the cost and the performance of *model adaptation*. The performance increases as the cost rises, gradually reaching convergence. This indicates a trade-off between the cost and the performance in *model adaptation*. When we pick an appropriate sample size and threshold, we can attain commendable performance at a comparatively modest expense.

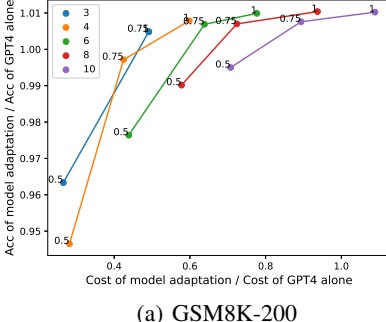

(a) GSM8K-200

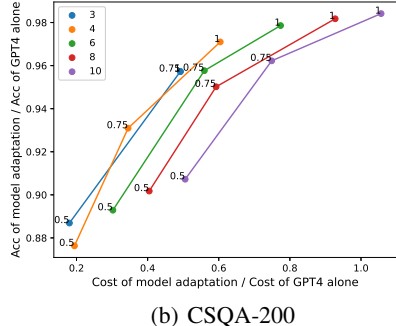

(b) CSQA-200

Figure 2: The balance of the performance and the cost in *model adaptation*. The horizontal axis represents the relative cost of *model adaptation* to the cost of using GPT4 alone. The vertical axis represents the relative accuracy of *model adaptation* to the accuracy of using GPT4 alone. Each color represents a value of sample size $N$, and the label of each point denotes a threshold $\theta$.

### 4.4 DETAILED ANALYSIS OF DECOMPOSITION GRANULARITY ADAPTATION

We delve deeper into examining the efficacy of adapting decomposition granularity. This is achieved through a comparative analysis of our method against its variants that fix decomposition granularity. In order to eliminate the impact of a multi-round solving strategy, we permit all non-adaptive decomposition prompts to address problems across multiple rounds, with a maximum limit of 3 rounds. Since the dataset CSQA and LLC do not require significant flexibility in terms of adjusting the decomposition granularity, we use only the six arithmetic reasoning datasets in this experiment. The results are reported in Table 3.

1) The *decomposition granularity adaptation* method (denoted as [$L_1$, $L_2$, $L_3$]), consistently outperforms other approaches on nearly all datasets. The exceptions are observed in the SingleEq and MultiArith datasets, where the problems are relatively easy, indicated by their high accuracies. In

these cases, problem decomposition may not be as crucial as it is for more challenging datasets. 2) The performances of non-adaptation variants vary among datasets. For example, on the GSM8K dataset, $[L_1, L_1, L_1]$ outperforms $[L_3, L_3, L_3]$, while on the SVAMP dataset, $[L_3, L_3, L_3]$ achieves better performance. This implies that it becomes a challenging task to achieve the best overall performance across all datasets when using fixed-granularity decomposition methods. Nonetheless, our approach tackles this challenge by dynamically adjusting the decomposition granularity for each problem. Consequently, we are able to attain the highest average performance across all datasets.

Table 3: Ablation experiment investigating the efficacy of *decomposition granularity adaptation*. L = L2M*, $L_1$ = (L2M*, $d_1$), $L_2$ = (L2M*, $d_2$), $L_3$ = (L2M*, $d_3$).

| Method | GSM8K | SVAMP | AQuA | AddSub | SingleEq | MultiArith | Average |
|---|---|---|---|---|---|---|---|
| [L, L, L] | 86.1 | 87.0 | 61.9 | 92.4 | 95.3 | 96.3 | 86.5 |
| $[L_1, L_1, L_1]$ | 87.0 | 86.1 | 62.0 | 91.9 | **95.6** | **98.7** | 86.9 |
| $[L_2, L_2, L_2]$ | 85.2 | 85.0 | 62.8 | 89.9 | 94.3 | 95.2 | 85.4 |
| $[L_3, L_3, L_3]$ | 85.5 | 88.3 | 62.4 | 92.2 | 95.4 | 95.7 | 86.6 |
| $[L_1, L_2, L_3]$ | **87.5** | **89.0** | **63.3** | **92.9** | **95.6** | 98.4 | **87.8** |
| Acc. Gain | +0.5 | +0.7 | +0.5 | +0.5 | +0.0 | -0.3 | +0.9 |

***Decomposition granularity adaptation* tailors decomposition granularity to problems with varied difficulties.** We conduct a performance comparison of different decomposition prompts when faced with increasingly challenging problems. The visualization is presented in Figure 3. 1) In Figure 3(a), it is evident that using a fixed-granularity decomposition method does not guarantee optimal performance across all problems. For example, for problems requiring fewer than 5 steps, the coarser-grained decomposition $[L_1, L_1, L_1]$ performs better than $[L_2, L_2, L_2]$. However, as the difficulty continues to increase, the finer-grained method $[L_2, L_2, L_2]$ exhibits superior performance. This demonstrates that problems of varying difficulty require different levels of decomposition. However, our adaptive decomposition method, denotes as $[L_1, L_2, L_3]$, consistently perform well across all settings, showcasing the advantage of *decomposition granularity adaptation*. 2) Figure 3(b) offers a further elucidation of the superior performance achieved by the adaptive method. The method dynamically selects the decomposition prompts for various problems. As the complexity of the problem escalates, it progressively enhances the utilization of finely-grained decomposition prompts, thereby resulting in an enhancement of overall performance.

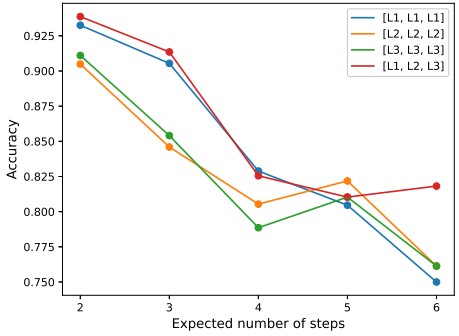

(a) Accuracy varies with the problem difficulty.

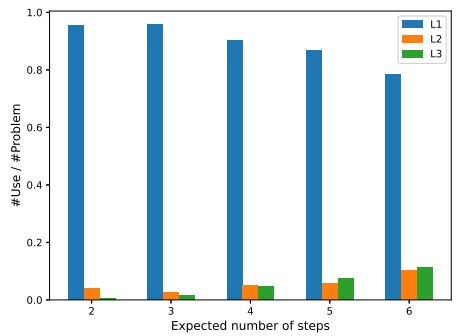

(b) The usage ratio of different decomposition prompts varies with the problem difficulty.

Figure 3: Analysis of decomposition granularity adaptation on GSM8K. The problem difficulty is measured by the number of expected solving steps, provided by the GSM8K dataset.

## 4.5 DETAILED ANALYSIS OF PROMPTING METHOD ADAPTATION

To further investigate the efficacy of *prompting method adaptation*, we compare our method with its non-adaptive variants that do not adjust prompting methods. Besides, we introduce an adaptive variant that reverses the order of solver list to study the order's impact. We provide two implementa-

tions of *prompting method adaptation*, with the solver list as: [CoT*, L2M*] and [ZeroCoT*, PS*]. The former implementation is focused on the domain of mathematical reasoning, whereas the latter targets broader and more general domains. We validate [CoT*, L2M*] on the arithmetic reasoning datasets, and [ZeroCoT*, PS*] on the commonsense reasoning dataset CSQA and the symbolic reasoning dataset LLC. The results are respectively reported in Table 4 and Table 5.

1) The optimal prompting method varies depending on the dataset. For instance, in Table 4, the method [C, C] performs better than [L, L] on three datasets but exhibits lower performance on the SVAMP dataset. 2) In Table 4 and Table 5, the adaptive methods (i.e., [C, L], [Z, P]) consistently outperform the non-adaptive variants. This suggests that *prompting method adaptation* can dynamically select a suitable prompting method for different problems across various datasets, resulting in an improved performance. 3) Besides, the sequence of various prompts (e.g., [C, L] and [L, C]) can impact the performance of *prompting method adaptation*. Empirically, it has been observed that shifting from simple and general prompting methods (e.g., CoT, ZeroCoT) to more intricate and specialized ones (e.g., L2M, PS) yields better results in overall.

Table 4: Ablation study of *prompting method adaptation* with the prompt list [C, L]. C = CoT*, L = L2M*.

| Method | GSM8K | SVAMP | AQuA | MultiArith |
|--------|-------|-------|------|------------|
| [C, C] | 85.5 | 82.4 | 62.8 | 98.3 |
| [L, L] | 82.9 | 86.3 | 58.3 | 95.7 |
| [C, L] | **85.7** | **87.0** | 65.0 | **98.7** |
| [L, C] | 85.4 | 85.6 | **66.5** | 98.3 |
| Acc. Gain | +0.2 | +0.7 | +3.7 | +0.4 |

Table 5: *Prompting method adaptation*. Z= ZeroCoT*, P= PS*.

| Method | CSQA | LLC |
|--------|------|-----|
| [Z, Z] | 74.3 | 84.6 |
| [P, P] | 73.7 | 81.8 |
| [Z, P] | **74.9** | **86.8** |
| [P, Z] | 74.5 | 83.6 |
| Acc. Gain | +0.6 | +2.2 |

***Prompting method adaptation* combines the advantages of different prompting methods.** As presented in Table 14 (Appendix A.10), we categorize all problems into four distinct groups based on the individual performance of CoT* and L2M*. We then measure the accuracy of the adaptive method [CoT*, L2M*] on each group, as well as the frequency of using CoT* and L2M* within the adaptive method. For the problems that both CoT* and L2M* successfully solve, we observed that [CoT*, L2M*] basically yields correct answers. Furthermore, for the subset of problems where either CoT* or L2M* succeeds while the other does not, [CoT*, L2M*] effectively address the majority (60%-70%) of them. These findings indicate that the adaptive approach effectively harnesses the complementary strengths of both prompting methods, leading to improved performance.

## 5 CONCLUSION AND DISCUSSION

We propose the Adaptive-Solver framework, a pioneering approach designed to dynamically tailor solving strategies for LLMs across diverse reasoning scenarios. Central to this framework are two modules: the initial evaluation module, which assesses the adequacy of a given solution, and, if refinement is necessary, activates the subsequent adaptation module. Herein, three adaptation strategies are leveraged: *model adaptation, prompting method adaptation*, and *decomposition granularity adaptation*. Our experimental findings underscore the efficacy of this framework, highlighting that adaptive prompting and decomposition granularity substantially augment the LLM's ability to pinpoint optimal methods and reasoning granularity, thereby markedly enhancing reasoning performance. Remarkably, the deployment of model adaptation brings a significant reduction in API costs, slashing them by up to 50%, all while sustaining or even amplifying performance. This innovative framework propels us into a promising direction in dynamic strategy selection for LLMs. Viewing from a higher point, every solver – be it model, prompting, decomposition, or augmented tools – can be regarded as a potential candidate in the component pool. The LLMs, armed with this framework, exhibit the flexibility to dynamically compose selected candidates, paving the way to optimal solution paths. Exploring future landscapes presents the exciting opportunity to widen the pool of problem solvers, paving the way for a broader spectrum of enhanced solutions. Take knowledge QA for example: if the model recognizes its limitations in addressing a particular query, it can wisely use retrieval-enhanced methods to navigate external knowledge for a optimal solution. This exploration not only highlights the versatile capabilities of LLMs but also uncovers routes for further investigations and innovations in adaptive problem-solving strategies.

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

# A APPENDIX

## A.1 DATASET DETAILS

We utilized the same dataset configuration as in (Wang et al., 2023a). In Table 6, you can find the dataset statistics.

Table 6: Details of the datasets. Math: arithmetic reasoning, CS: commonsense reasoning, Sym.: symbolic reasoning.

| Dataset | Domain | # Samples | Ave. words | Anaswer |
|---------|--------|-----------|------------|---------|
| GSM8K | Math | 1319 | 46.9 | Number |
| SVAMP | Math | 1000 | 31.8 | Number |
| AQUA | Math | 254 | 51.9 | Option |
| AddSub | Math | 395 | 31.5 | Number |
| SingleEq | Math | 508 | 27.4 | Number |
| CSQA | CS | 1221 | 27.8 | Option |
| LLC | Sym. | 500 | 15.0 | String |

## A.2 IMPLEMENTATION DETAILS

We use GPT-3.5-turbo-0301 as the LLM model unless otherwise stated. We set the temperature as 0 for the greedy decoding strategy used in single-solution baselines and PHP, while 0.7 for the methods with self-consistency strategy. When using self-consistency, we set the sampling size to 3 unless otherwise specified. Our experiments were conducted from June 2023 to September 2023. During this period, the API prices for GPT-3.5-turbo were {"input": $0.0015 / 1K tokens, "output": $0.002 / 1K tokens}, while for GPT-4, the corresponding rates were {"input": $0.03 / 1K tokens, "output": $0.06 / 1K tokens}.

## A.3 PERFORMANCE COMPARISON ON THE CSQA AND THE LLC

Table 7 demonstrates the performance comparison of our adaptive method AS-P with zero-shot and four-shot prompting baselines on the CSQA and LLC datasets. See the specific analysis in 4.2.

Table 7: Performance comparison on the CSQA and LLC datasets. The best results are boldfaced. AS-P uses the solver list: [ZeroCoT*, PS*].

| | Method | CSQA | LLC | Average |
|-----------|---------|------|------|---------|
| 4-shot | CoT | 73.1 | 81.0 | 77.1 |
| | CoT_SC | 72.1 | 81.0 | 76.6 |
| zero-shot | ZeroCoT | 70.4 | 70.6 | 70.5 |
| | PS | 69.8 | 63.6 | 66.7 |
| | AS-P (Ours) | **74.9** | **86.8** | **80.9** |

## A.4 EFFICIENCY ANALYSIS OF THE ADAPTIVE METHODS.

We will analyze the adaptive methods' efficiency from two aspects: the average number of solving rounds and the average inference time.

Table 8 provides statistics on the average number of solving iterations required by various adaptive methods on the arithmetic reasoning datasets. While the adaptive methods employ multi-round solving strategy, the actual average number of iterations hasn't shown a significant increase (averaging

less than 1.5 iterations across multiple datasets), thus resulting in only a marginal increase in cost compared with the methods that adopt single-round solving strategy.

Table 8: Average number of solving rounds of the adaptative methods. AS-P uses the solver list [COT*, L2M*], AS-D uses the solver list [(L2M*, $d_1$), (L2M*, $d_2$), (L2M*, $d_3$)], AS-PD uses the solver list [CoT*, (L2M*, $d_1$), (L2M*, $d_2$), (L2M*, $d_3$)].

| Method | GSM8K | SVAMP | AQuA | AddSub | SingleEq | MultiArith | Average |
|---|---|---|---|---|---|---|---|
| AS-P | 1.26 | 1.22 | 1.54 | 1.10 | 1.06 | 1.03 | 1.20 |
| AS-D | 1.46 | 1.38 | 2.12 | 1.17 | 1.10 | 1.08 | 1.38 |
| AS-PD | 1.59 | 1.49 | 2.27 | 1.14 | 1.11 | 1.06 | 1.44 |

Table 9 and Table 10 present the average inference time per problem associated with various prompting methods. We observe that, akin to the number of solving rounds, there is not a significant increase in inference time. Specifically, the average inference time of AS-P (AS-D and AS-PD) is no more than 1.6 times that of COT* (L2M*). The average inference time of AS-M is no more than 1.2 times that of GPT4. The rationale behind this is that the majority of problems are resolved by the initial solver, with the subsequent solvers only being invoked in a few necessary cases.

Table 9: Inference time (seconds) change of different types of adaptation. AS-P uses the solver list [COT*, L2M*], AS-D uses the solver list [(L2M*, $d_1$), (L2M*, $d_2$), (L2M*, $d_3$)].

| Method | GSM8K | SVAMP | AQuA |
|---|---|---|---|
| COT* | 4.121 | 1.775 | 6.629 |
| L2M* | 6.473 | 2.677 | 8.801 |
| AS-P | 6.264 | 2.447 | 7.771 |
| AS-D | 10.319 | 3.582 | 11.248 |
| AS-PD | 9.118 | 3.542 | 13.576 |

Table 10: Inference time (seconds) change of Model Adaptation. AS-M uses the solver list [GPT3.5*, GPT4].

| Method | GSM8K-200 | SVAMP-200 | AQuA |
|---|---|---|---|
| GPT3.5* | 7.804 | 6.197 | 5.905 |
| GP4 | 8.595 | 9.783 | 13.94 |
| AS-M | 9.722 | 9.818 | 4.818 |

### A.5 IMPACT OF VARYING SAMPLING TEMPERATURE ON PERFORMANCE OF DIFFERENT ADAPTATIONS

We examine how the performances of the three types of adaptation vary with the sampling temperature. The results are presented in Table 11. Across all three adaptation types, as the temperature increases the performances consistently get higher. This phenomenon can be attributed to the fact that a higher sampling temperature enhances answer diversity. This potentially improves the precision of consistency checks in classification and gives more chances to answer a question.

### A.6 ANALYSIS OF EVALUATION ABILITY OF CONSISTENCY CHECK

The evaluation module, implemented through the consistency check method, can be viewed as a binary classification task focused on determining the correctness of the current solution. To measure the evaluation ability of consistency check, we can utilize ROC (Receiver Operating Characteristic)

Table 11: Effect of varying temperature on performance (i.e., Accuracy) of different adaptations on the GSM8K dataset. AS-P uses the solver list [COT*, L2M*], AS-D uses the solver list [(L2M*, $d_1$), (L2M*, $d_2$), (L2M*, $d_3$)], AS-M uses the solver list [GPT3.5*, GPT4]. T: Temperture.

| Method | T=0.4 | T=0.7 | T=1.0 |
|--------|-------|-------|-------|
| AS-P   | 84.2  | 85.7  | 85.7  |
| AS-D   | 85.9  | 87.5  | 87.2  |
| AS-M   | 92.5  | 93.5  | 95.5  |

curve and AUC (Area Under the Curve) as metrics. We present the ROC curve of the consistency check (sample size = 3) for four different prompting methods at a range of temperatures {0.4, 0.7, 1.0}. The experiment was conducted with the GPT-3.5 model, on the GSM8K dataset. As shown in Figure 4, as the temperature increases, the performances (i.e., AUC) of the consistency check also improve but converge when temperature (at 0.7) gets high enough.

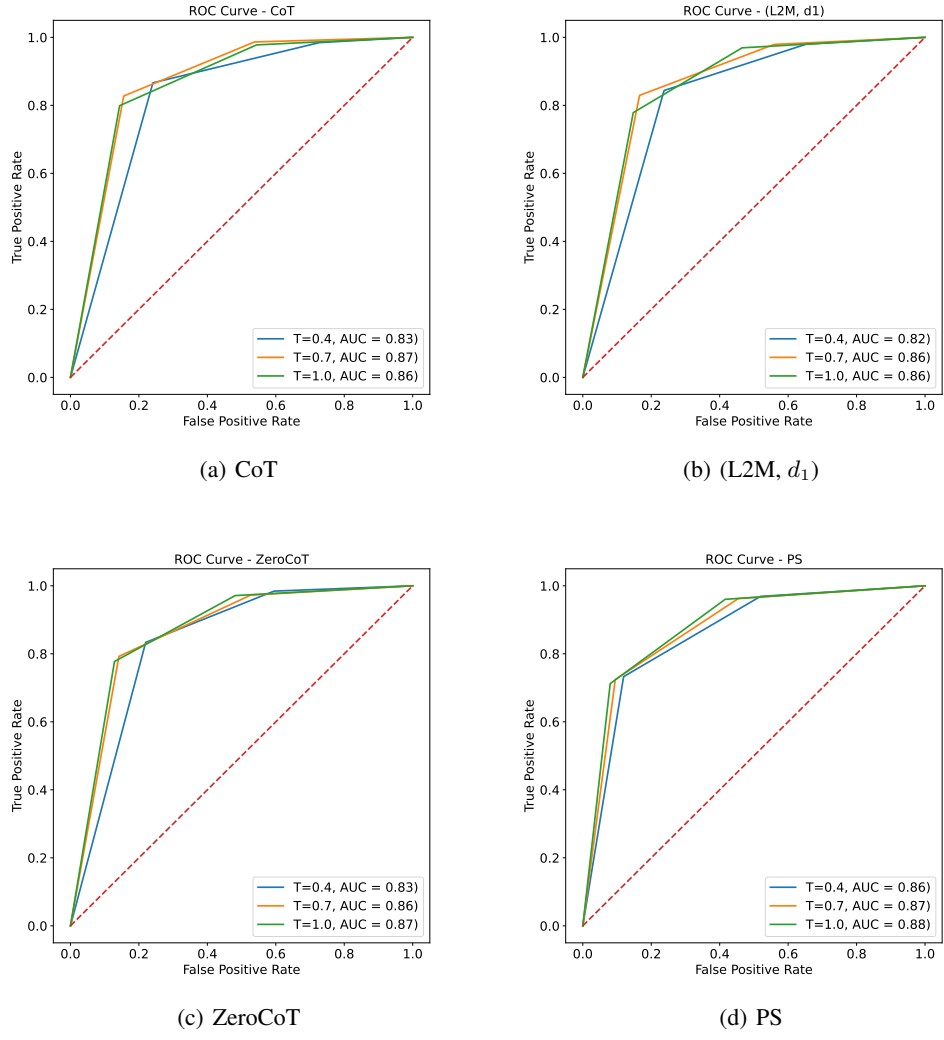

Figure 4: The ROC curve of the consistency check for prompting CoT, (L2M, $d_1$), ZeroCoT and PS.

## A.7 DISCUSSION OF DETERMINING AN OPTIMAL SOLVER LIST FOR A GIVEN DATASET

We provide two approaches to determine an optimal cascade per budget for a given dataset. 1) The first method is a broad approach applicable for all the types of adaptation. We can sample a small subset of the dataset as a validation set and try different permutations and combinations of solvers to decide which solvers to include and the optimal order in the solver list that meet the budget limit. 2) The second method specifically targets model adaptation, with the goal of maintaining performance while minimizing costs. Typically, as the model's capabilities increase, so does the associated cost. Hence, a pragmatic strategy involves constructing the solver list by arranging them from weaker to stronger based on the model's strength.

## A.8 ADDITIONAL RESULTS OF *model adaptation*

From table 12, we can observe that: 1) GPT3.5 exhibits a significant advantage over GLM2*, with a difference of approximately 30-60%. However, it's worth noting that GPT3.5 is a closed-source paid model. In contrast, GLM2 is an open-source, locally deployable model, so we do not consider the cost of GLM2 for simplicity. 2) Although [GLM2*, GLM2*] enhances the performance of GLM2* by solving problems over multiple rounds, a substantial disparity persists between [GLM2*, GLM2*] and [GLM2*, GPT3.5]. This underscores the inherent limitations of weaker models. 3) Given the substantial performance gap between GLM2* and GPT3.5, it remains challenging for [GLM2*, GPT3.5] to match the performance of GPT3.5. Nonetheless, by allowing a performance decrease of no more than 2%, we can still achieve a cost (i.e., the number of calling of GPT3.5) reduction, roughly ranging from 13% to 30%.

Table 12: Model adaptation implemented with the model list [GLM2*, GPT3.5]. #Call means the number of calling GPT3.5. We calculate the cost with the calling number of GPT3.5.

| LLM Model | GSM8K-200 | | SVAMP-200 | | AQuA | |
|---|---|---|---|---|---|---|
| | ACC | #Call | ACC | #Call | ACC | #Call |
| GLM2* | 35.5 | 0 | 48.5 | 0 | 25.2 | 0 |
| [GLM2*,GLM2*] | 36.0 | 0 | 49.0 | 0 | 30.7 | 0 |
| GPT3.5 | 83.5 | 200 | 78.5 | 200 | 63.4 | 254 |
| [GLM2*,GPT3.5] | 81.9 | 153 | 76.8 | 139 | 61.6 | 221 |
| Acc. Gain / Saved #Call | -1.6 / 23.5% | | -1.7 / 30.4% | | -1.8 / 12.9% | |

## A.9 ADDITIONAL RESULTS OF *decomposition granularity adaptation*

Table 13 demonstrates the average number of sub-problems on different datasets obtained by using various decomposition prompting methods. We observe that finer-grained decomposition prompt indeed leads to a greater number of subproblems on average on the same dataset. This validates the effectiveness of controlling the granularity in the actual problem decomposition by modulating the granularity in the exemplars.

Table 13: Average number of sub-problems of various decomposition prompting methods.

| Method | GSM8K | SVAMP | MultiArith | AddSub | SingleEq | AQuA | Average |
|---|---|---|---|---|---|---|---|
| L2M | 3.61 | 2.76 | 2.80 | 2.51 | 2.63 | 3.08 | 2.90 |
| (L2M, $d_1$) | 2.60 | 1.88 | 2.06 | 1.73 | 1.77 | 2.19 | 2.04 |
| (L2M, $d_2$) | 3.6 | 2.76 | 2.73 | 2.44 | 2.54 | 2.74 | 2.80 |
| (L2M, $d_3$) | 4.46 | 3.56 | 3.51 | 2.85 | 3.15 | 3.57 | 3.52 |

A.10 ADDITIONAL RESULTS OF *prompting method adaptation*

Table 14 presents the statistical results for analysis of *prompting method adaptation*. See the specific analysis in 4.5.

Table 14: Analysis of prompting method adaptation. CoT* ✓and L2M* ✗ means the problems that CoT* solves successfully while L2M* fails.

| Dataset | CoT* | L2M* | # problems | # correct problems by [CoT*, L2M*] | CoT* usage count | L2M* usage count |
|---|---|---|---|---|---|---|
| GSM8K | ✓ | ✓ | 995 | 984 (98.9%) | 884 | 11 |
| | ✓ | ✗ | 123 | 76 (61.8%) | 62 (50.4%) | 61 |
| | ✗ | ✓ | 84 | 56 (66.7%) | 31 | 53 (63.1%) |
| | ✗ | ✗ | 117 | 25 (21.4%) | 50 | 67 |
| SVAMP | ✓ | ✓ | 762 | 755 (99.1%) | 692 | 70 |
| | ✓ | ✗ | 51 | 34 (66.7%) | 31 (60.8%) | 20 |
| | ✗ | ✓ | 94 | 65 (69.1%) | 34 | 60 (63.8%) |
| | ✗ | ✗ | 93 | 15 (16.1%) | 45 | 48 |

A.11 APPROACH FOR CONSTRUCTING (L2M, $d_i$)'S PROMPT IN *decomposition granularity adaptation*

To illustrate, consider the following example question: *Cappuccinos cost $2, iced teas cost $3, cafe lattes cost $1.5 and espressos cost $1 each. Sandy orders some drinks for herself and some friends. She orders three cappuccinos, two iced teas, two cafe lattes, and two espressos. How much change does she receive back for a twenty-dollar bill?*

L2M does not control the decomposition granularity deliberately and its decomposition for the example question is as follows: *1. How much did the cappuccinos cost in total? 2. How much did the iced teas cost in total? 3. How much did the cafe lattes cost in total? 4. How much did the espressos cost in total? 5. How much did Sandy spend on drinks? 6. How much change does she receive back for a twenty-dollar bill?*

To construct L2M's variants, we first decompose the question hierarchically, as shown in Figure 5.

1) First, we extract the problem and sub-problems from the first layer of decomposition. Then, serialize them from bottom to top to obtain the sequence of sub-problems in (L2M, $d_1$)'s prompt: *1. How much did Sandy spend on drinks? 2. How much change does she receive back for a twenty-dollar bill?*

2) Similarly, we extract the problem and sub-problems from the first two layers of decomposition and then serialize them to obtain the sequence of sub-problems in (L2M, $d_2$)'s prompt: *1. How much did the cappuccinos cost in total? 2. How much did the iced teas cost in total? 3. How much did the cafe lattes cost in total? 4. How much did the espressos cost in total? 5. How much did Sandy spend on drinks? 6. How much change does she receive back for a twenty-dollar bill?*

3) Likewise, we extract the problem and sub-problems from the three layers of decomposition and serialize them to obtain the sequence of sub-problems in (L2M, $d_3$)'s prompt: *1. How many cappuccinos did Sandy order? 2. How much did the cappuccinos cost in total? 3. How many iced teas did Sandy order? 4. How much did the iced teas cost in total? 5. How many cafe lattes did Sandy order? 6. How much did the cafe lattes cost in total? 7. How many espressos did Sandy order? 8. How much did the espressos cost in total? 9. How much did Sandy spend on all drinks in total? 10. How much change does she receive back for a twenty-dollar bill?*

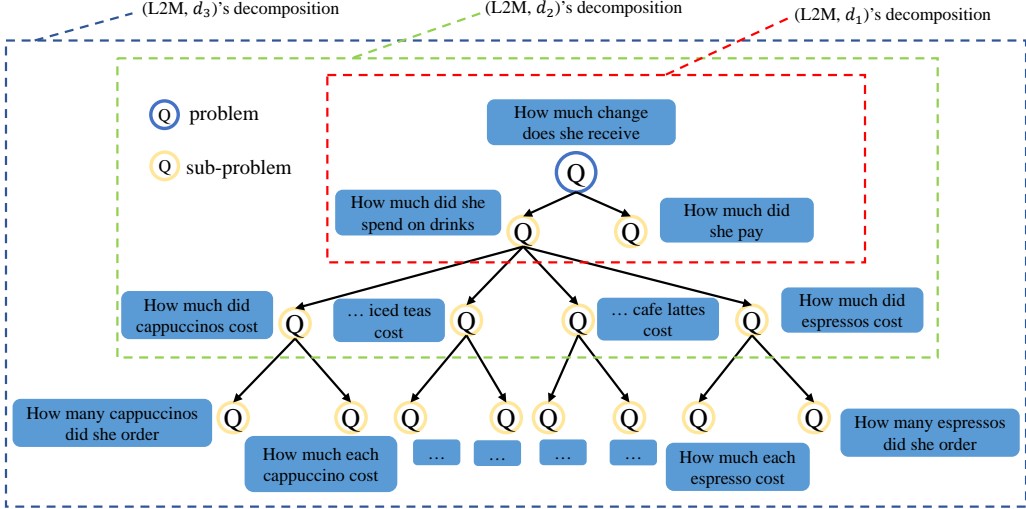

Figure 5: Illustration of hierarchical decomposition.

## A.12 FULL SETS OF PROMPTS

We present all the prompts used in this work. For all the prompts, if we do not detect "answer is" in the response, we concatenate the question, the response and "Therefore, the answer is" to call API once gain, to obtain a short response containing the answer.

### A.12.1 ZERO-SHOT-COT (ZEROCOT)

> **Prompt for all the datasets**:
> Q: {question}
> A: Let's think step by step.

### A.12.2 PLAN-AND-SOLVE (PS)

> **Prompt for all the arithmetic reasoning datasets**:
> Q: {question}
> A: Let's first understand the problem, extract relevant variables and their corresponding numerals, and make and devise a complete plan. Then, let's carry out the plan, calculate intermediate variables (pay attention to correct numerical calculation and commonsense), solve the problem step by step, and show the answer.

> **Prompt for the commonsense reasoning dataset CSQA**:
> Q: {question}
> A: Let's first prepare relevant information and make a plan. Then, let's answer the question step by step (pay attention to commonsense and logical coherence).

> **Prompt for the symbolic reasoning dataset LLC**:
> Q: {question}
> A: Let's devise a plan and solve the problem step by step.

### A.12.3 CHAIN-OF-THOUGHT (COT) & COT_SC

---

**Four-shot exemplars for all the mathematical reasoning datasets excluding the AQuA dataset**:

Q: Four years ago, Kody was only half as old as Mohamed. If Mohamed is currently twice 30 years old, how old is Kody?

A: We were told that Mohamed is currently twice 30 years old, so he is currently 30 * 2 = 60 years old. That means that four years ago he must have been 60 - 4 = 56 years old. Four years ago, Kody was half as old as Mohamed, so Kody must have been 56 / 2 = 28 years old then. Since Kody was 28 years old four years ago, she must now be 28 + 4 = 32 years old. The answer is 32.

Q: Carla bought 2 bags of mini peanut butter cups on clearance. Each bag was $6.00 but was 75% off. How much did she spend on 2 bags of candy?

A: Each bag was $6.00 but was 75% off. So each bag cost $6.00 * (1 - 0.75) = $6.00 * 0.25 = $1.50. Carla bought 2 bags. So she spent $1.50 * 2 = $3.00. The answer is 3.

Q: If Pam is currently twice as young as Rena is, and in 10 years Rena will be 5 years older than her, how old is Pam now?

A: Since Rena will be 5 years older than Pam in 10 years, she must be 5 years older than Pam now as well. If Pam is currently twice as young as Rena, that means that Rena is currently twice as old as Pam is. So if P stands for Pam's age now and R stands for Rena's age now, then we know that R = 2 * P And since Rena is 5 years older than Pam now, we know that R = P + 5. By substitution, we have P + 5 = 2 * P, which means that P = 5. The answer is 5.

Q: Cappuccinos cost $2, iced teas cost $3, cafe lattes cost $1.5 and espressos cost $1 each. Sandy orders some drinks for herself and some friends. She orders three cappuccinos, two iced teas, two cafe lattes, and two espressos. How much change does she receive back for a twenty-dollar bill?

A: Sandy ordered three cappuccinos, which cost $2 each, so she spent $2 * 3 = $6 on cappuccinos. She ordered two iced teas, which cost $3 each, so she spent $3 * 2 = $6 dollars on ice teas. She ordered two cafe lattes, which cost $1.5 each, so she spent $1.5 * 2 = $3 on cafe lattes. She ordered two espressos, which cost $1 each, so she spent $1 * 2 = $2 on espressos. So altogether, Sandy spent $6 + $6 + $3 + $2 = $17 on drinks, which means that sandy will get $20 - $17 = $3 as change. The answer is 3.

Q: {question}

A:

---

**Four-shot exemplars for the AQuA dataset**:

Q: Four years ago, Kody was only half as old as Mohamed. If Mohamed is currently twice 30 years old, how old is Kody? Answer Choices: (a) 33 (b) 32 (c) 16 (d) 20

A: We were told that Mohamed is currently twice 30 years old, so he is currently 30 * 2 = 60 years old. That means that four years ago he must have been 60 - 4 = 56 years old. Four years ago, Kody was half as old as Mohamed, so Kody must have been 56 / 2 = 28 years old then. Since Kody was 28 years old four years ago, she must now be 28 + 4 = 32 years old. The answer is 32. Therefore, the answer is (b).

Q: Carla bought 2 bags of mini peanut butter cups on clearance. Each bag was $6.00 but was 75% off. How much did she spend on 2 bags of candy? Answer Choices: (a) 1.5 (b) 6 (c) 5 (d) 3

A: Each bag was $6.00 but was 75% off. So each bag cost $6.00 * (1 - 0.75) = $6.00 * 0.25 = $1.50. Carla bought 2 bags. So she spent $1.50 * 2 = $3.00. The answer is 3. Therefore, the answer is (d).

Q: If Pam is currently twice as young as Rena is, and in 10 years Rena will be 5 years older than her, how old is Pam now? Answer Choices: (a) 5 (b) 10 (c) 15 (d) 20 (e) 30

A: Since Rena will be 5 years older than Pam in 10 years, she must be 5 years older than Pam now as well. If Pam is currently twice as young as Rena, that means that Rena is currently twice as old as Pam is. So if P stands for Pam's age now and R stands for Rena's age now, then we know that R = 2 * P And since Rena is 5 years older than Pam now, we know that R = P + 5. By substitution, we have P + 5 = 2 * P, which means that P = 5. The answer is 5. Therefore, the answer is (a).

Q: Cappuccinos cost $2, iced teas cost $3, cafe lattes cost $1.5 and espressos cost $1 each. Sandy orders some drinks for herself and some friends. She orders three cappuccinos, two iced teas, two cafe lattes, and two espressos. How much change does she receive back for a twenty-dollar bill? Answer Choices: (a) 20 (b) 17 (c) 3 (d) 1

A: Sandy ordered three cappuccinos, which cost $2 each, so she spent $2 * 3 = $6 on cappuccinos. She ordered two iced teas, which cost $3 each, so she spent $3 * 2 = $6 dollars on ice teas. She ordered two cafe lattes, which cost $1.5 each, so she spent $1.5 * 2 = $3 on cafe lattes. She ordered two espressos, which cost $1 each, so she spent $1 * 2 = $2 on espressos. So altogether, Sandy spent $6 + $6 + $3 + $2 = $17 on drinks, which means that sandy will get $20 - $17 = $3 as change. The answer is 3. Therefore, the answer is (c).

Q: {question}

A:

---

**Four-shot exemplars for the commonsense reasoning dataset CSQA**:

Q: What do people use to absorb extra ink from a fountain pen? Answer Choices: (a) shirt pocket (b) calligrapher's hand (c) inkwell (d) desk drawer (e) blotter

A: The answer must be an item that can absorb ink. Of the above choices, only blotters are used to absorb ink. So the answer is (e).

Q: What home entertainment equipment requires cable? Answer Choices: (a) radio shack (b) substation (c) television (d) cabinet

A: The answer must require cable. Of the above choices, only television requires cable. So the answer is (c).

Q: The fox walked from the city into the forest, what was it looking for? Answer Choices: (a) pretty flowers (b) hen house (c) natural habitat (d) storybook

A: The answer must be something in the forest. Of the above choices, only natural habitat is in the forest. So the answer is (b).

Q: Sammy wanted to go to where the people were. Where might he go? Answer Choices: (a) populated areas (b) race track (c) desert (d) apartment (e) roadblock

A: The answer must be a place with a lot of people. Of the above choices, only populated areas have a lot of people. So the answer is (a).

Q: question

A:

---

**Four-shot exemplars for the symbolic reasoning dataset LLC**:

Q: Take the last letters of the words in "Elon Musk" and concatenate them.

A: The last letter of "Elon" is "n". The last letter of "Musk" is "k". Concatenating them is "nk". The answer is "nk".

Q: Take the last letters of the words in "Larry Page" and concatenate them.

A: The last letter of "Larry" is "y". The last letter of "Page" is "e". Concatenating them is "ye". The answer is "ye".

Q: Take the last letters of the words in "Sergey Brin" and concatenate them.

A: The last letter of "Sergey" is "y". The last letter of "Brin" is "n". Concatenating them is "yn". The answer is "yn".

Q: Take the last letters of the words in "Bill Gates" and concatenate them.

A: The last letter of "Bill" is "l". The last letter of "Gates" is "s". Concatenating them is "ls". The answer is "ls".

Q: question

A:

### A.12.4 LEAST-TO-MOST (L2M)

> **Four-shot exemplars for all the mathematical reasoning datasets excluding the AQuA dataset**:
> Q: Four years ago, Kody was only half as old as Mohamed. If Mohamed is currently twice 30 years old, how old is Kody?
> A: Let's break down this problem: 1. How old is Mohamed currently? 2. How old was Mohamed four years ago? 3. How old was Kody four years ago? 4. How old is Kody?
> 1. We were told that Mohamed is currently twice 30 years old, so he is currently 30 * 2 = 60 years old. The answer is 60.
> 2. That means that four years ago he must have been 60 - 4 = 56 years old. The answer is 56.
> 3. Four years ago, Kody was half as old as Mohamed, so Kody must have been 56 / 2 = 28 years old then. The answer is 28.
> 4. Since Kody was 28 years old four years ago, she must now be 28 + 4 = 32 years old. The answer is 32.
> Q: Carla bought 2 bags of mini peanut butter cups on clearance. Each bag was $6.00 but was 75% off. How much did she spend on 2 bags of candy?
> A: Let's break down this problem: 1. How much did she spend on 2 bags of candy?
> 1. Each bag was $6.00 but was 75% off. So each bag cost $6.00 * (1 - 0.75) = $6.00 * 0.25 = $1.50. Carla bought 2 bags. So she spent $1.50 * 2 = $3.00. The answer is 3.
> Q: If Pam is currently twice as young as Rena is, and in 10 years Rena will be 5 years older than her, how old is Pam now?
> A: Let's break down this problem: 1. How much older is Rena than Pam currently? 2. How old is Pam now?
> 1. Since Rena will be 5 years older than Pam in 10 years, she must be 5 years older than Pam now as well. The answer is 5.
> 2. If Pam is currently twice as young as Rena, that means that Rena is currently twice as old as Pam is. So if P stands for Pam's age now and R stands for Rena's age now, then we know that R = 2 * P And since Rena is 5 years older than Pam now, we know that R = P + 5. By substitution, we have P + 5 = 2 * P, which means that P = 5. The answer is 5.
> Q: Cappuccinos cost $2, iced teas cost $3, cafe lattes cost $1.5 and espressos cost $1 each. Sandy orders some drinks for herself and some friends. She orders three cappuccinos, two iced teas, two cafe lattes, and two espressos. How much change does she receive back for a twenty-dollar bill?
> A: Let's break down this problem: 1. How much did the cappuccinos cost in total? 2. How much did the iced teas cost in total? 3. How much did the cafe lattes cost in total? 4. How much did the espressos cost in total? 5. How much did Sandy spend on drinks? 6. How much change does she receive back for a twenty-dollar bill?
> 1. Sandy ordered three cappuccinos, which cost $2 each, so she spent $2 * 3 = $6 on cappuccinos. The answer is 6.
> 2. She ordered two iced teas, which cost $3 each, so she spent $3 * 2 = $6 dollars on ice teas. The answer is 6.
> 3. She ordered two cafe lattes, which cost $1.5 each, so she spent $1.5 * 2 = $3 on cafe lattes. The answer is 3.
> 4. She ordered two espressos, which cost $1 each, so she spent $1 * 2 = $2 on espressos. The answer is 2.
> 5. So altogether, Sandy spent $6 + $6 + $3 + $2 = $17 on drinks. The answer is 17.
> 6. Sandy will get $20 - $17 = $3 as change. The answer is 3.
> Q: {question}
> A: Let's break down this problem:

**Four-shot exemplars for the AQuA dataset**:

Q: Four years ago, Kody was only half as old as Mohamed. If Mohamed is currently twice 30 years old, how old is Kody? Answer Choices: (a) 33 (b) 32 (c) 16 (d) 20

A: Let's break down this problem: 1. How old is Mohamed currently? 2. How old was Mohamed four years ago? 3. How old was Kody four years ago? 4. How old is Kody?

1. We were told that Mohamed is currently twice 30 years old, so he is currently 30 * 2 = 60 years old. The answer is 60.

2. That means that four years ago he must have been 60 - 4 = 56 years old. The answer is 56.

3. Four years ago, Kody was half as old as Mohamed, so Kody must have been 56 / 2 = 28 years old then. The answer is 28.

4. Since Kody was 28 years old four years ago, she must now be 28 + 4 = 32 years old. The answer is 32. Therefore, the answer is (b).

Q: Carla bought 2 bags of mini peanut butter cups on clearance. Each bag was $6.00 but was 75% off. How much did she spend on 2 bags of candy? Answer Choices: (a) 1.5 (b) 6 (c) 5 (d) 3

A: Let's break down this problem: 1. How much did she spend on 2 bags of candy?

1. Each bag was $6.00 but was 75% off. So each bag cost $6.00 * (1 - 0.75) = $6.00 * 0.25 = $1.50. Carla bought 2 bags. So she spent $1.50 * 2 = $3.00. The answer is 3. Therefore, the answer is (d).

Q: If Pam is currently twice as young as Rena is, and in 10 years Rena will be 5 years older than her, how old is Pam now? Answer Choices: (a) 5 (b) 10 (c) 15 (d) 20 (e) 30

A: Let's break down this problem: 1. How much older is Rena than Pam currently? 2. How old is Pam now?

1. Since Rena will be 5 years older than Pam in 10 years, she must be 5 years older than Pam now as well. The answer is 5.

2. If Pam is currently twice as young as Rena, that means that Rena is currently twice as old as Pam is. So if P stands for Pam's age now and R stands for Rena's age now, then we know that R = 2 * P And since Rena is 5 years older than Pam now, we know that R = P + 5. By substitution, we have P + 5 = 2 * P, which means that P = 5. The answer is 5. Therefore, the answer is (a).

Q: Cappuccinos cost $2, iced teas cost $3, cafe lattes cost $1.5 and espressos cost $1 each. Sandy orders some drinks for herself and some friends. She orders three cappuccinos, two iced teas, two cafe lattes, and two espressos. How much change does she receive back for a twenty-dollar bill? Answer Choices: (a) 20 (b) 17 (c) 3 (d) 1

A: Let's break down this problem: 1. How much did the cappuccinos cost in total? 2. How much did the iced teas cost in total? 3. How much did the cafe lattes cost in total? 4. How much did the espressos cost in total? 5. How much did Sandy spend on drinks? 6. How much change does she receive back for a twenty-dollar bill?

1. Sandy ordered three cappuccinos, which cost $2 each, so she spent $2 * 3 = $6 on cappuccinos. The answer is 6.

2. She ordered two iced teas, which cost $3 each, so she spent $3 * 2 = $6 dollars on ice teas. The answer is 6.

3. She ordered two cafe lattes, which cost $1.5 each, so she spent $1.5 * 2 = $3 on cafe lattes. The answer is 3.

4. She ordered two espressos, which cost $1 each, so she spent $1 * 2 = $2 on espressos. The answer is 2.

5. So altogether, Sandy spent $6 + $6 + $3 + $2 = $17 on drinks. The answer is 17.

6. Sandy will get $20 - $17 = $3 as change. The answer is 3. Therefore, the answer is (c).

Q: {question}

A: Let's break down this problem:

### A.12.5  L2M'S VARIANTS FOR DECOMPOSITION GRANULARITY ADAPTATION

The following three prompts mainly differ from the decomposition granularity. For example, facing the same problem, the prompt (L2M, $d_1$) may break it down into 2-3 sub-questions, the prompt (L2M, $d_2$) may decompose it into 4-5 sub-questions, and the prompt (L2M, $d_3$) may decompose it into 6-8 sub-questions. In addition, the difference between them and L2M lies in: L2M lacks precise control over decomposition granularity in its demonstrations, leading to a blend of various granularities. Conversely, in the demonstrations of these variants, the decomposition granularity is either coarse, medium, or fine, depending on the specific variant.

**(1) The prompts of (L2M, $d_1$)**

**Four-shot exemplars for all the mathematical reasoning datasets excluding the AQuA dataset**:

Q: Four years ago, Kody was only half as old as Mohamed. If Mohamed is currently twice 30 years old, how old is Kody?

A: Let's break down this problem: 1. How old was Kody four years ago? 2. How old is Kody?

1. We were told that Mohamed is currently twice 30 years old, so he is currently 30 * 2 = 60 years old. That means that four years ago he must have been 60 - 4 = 56 years old. Four years ago, Kody was half as old as Mohamed, so Kody must have been 56 / 2 = 28 years old then. The answer is 28.

2. Since Kody was 28 years old four years ago, she must now be 28 + 4 = 32 years old. The answer is 32.

Q: Carla bought 2 bags of mini peanut butter cups on clearance. Each bag was $6.00 but was 75% off. How much did she spend on 2 bags of candy?

A: Let's break down this problem: 1. How much did she spend on 2 bags of candy?

1. Each bag was $6.00 but was 75% off. So each bag cost $6.00 * (1 - 0.75) = $6.00 * 0.25 = $1.50. Carla bought 2 bags. So she spent $1.50 * 2 = $3.00. The answer is 3.

Q: If Pam is currently twice as young as Rena is, and in 10 years Rena will be 5 years older than her, how old is Pam now?

A: Let's break down this problem: 1. How much older is Rena than Pam currently? 2. How old is Pam now?

1. Since Rena will be 5 years older than Pam in 10 years, she must be 5 years older than Pam now as well. The answer is 5.

2. If Pam is currently twice as young as Rena, that means that Rena is currently twice as old as Pam is. So if P stands for Pam's age now and R stands for Rena's age now, then we know that R = 2 * P And since Rena is 5 years older than Pam now, we know that R = P + 5. By substitution, we have P + 5 = 2 * P, which means that P = 5. The answer is 5.

Q: Cappuccinos cost $2, iced teas cost $3, cafe lattes cost $1.5 and espressos cost $1 each. Sandy orders some drinks for herself and some friends. She orders three cappuccinos, two iced teas, two cafe lattes, and two espressos. How much change does she receive back for a twenty-dollar bill?

A: Let's break down this problem: 1. How much did Sandy spend on drinks? 2. How much change does she receive back for a twenty-dollar bill?

1. Sandy ordered three cappuccinos, which cost $2 each, so she spent $2 * 3 = $6 on cappuccinos. She ordered two iced teas, which cost $3 each, so she spent $3 * 2 = $6 dollars on ice teas. She ordered two cafe lattes, which cost $1.5 each, so she spent $1.5 * 2 = $3 on cafe lattes. She ordered two espressos, which cost $1 each, so she spent $1 * 2 = $2 on espressos. So altogether, Sandy spent $6 + $6 + $3 + $2 = $17 on drinks. The answer is 17.

2. Sandy will get $20 - $17 = $3 as change. The answer is 3.

Q: {question}

A: Let's break down this problem:

**Four-shot exemplars for the AQuA dataset**:

Q: Four years ago, Kody was only half as old as Mohamed. If Mohamed is currently twice 30 years old, how old is Kody? Answer Choices: (a) 33 (b) 32 (c) 16 (d) 20

A: Let's break down this problem: 1. How old was Kody four years ago? 2. How old is Kody?

1. We were told that Mohamed is currently twice 30 years old, so he is currently 30 * 2 = 60 years old. That means that four years ago he must have been 60 - 4 = 56 years old. Four years ago, Kody was half as old as Mohamed, so Kody must have been 56 / 2 = 28 years old then. The answer is 28.

2. Since Kody was 28 years old four years ago, she must now be 28 + 4 = 32 years old. The answer is 32. Therefore, the answer is (b).

Q: Carla bought 2 bags of mini peanut butter cups on clearance. Each bag was $6.00 but was 75% off. How much did she spend on 2 bags of candy? Answer Choices: (a) 1.5 (b) 6 (c) 5 (d) 3

A: Let's break down this problem: 1. How much did she spend on 2 bags of candy?

1. Each bag was $6.00 but was 75% off. So each bag cost $6.00 * (1 - 0.75) = $6.00 * 0.25 = $1.50. Carla bought 2 bags. So she spent $1.50 * 2 = $3.00. The answer is 3. Therefore, the answer is (d).

Q: If Pam is currently twice as young as Rena is, and in 10 years Rena will be 5 years older than her, how old is Pam now? Answer Choices: (a) 5 (b) 10 (c) 15 (d) 20 (e) 30 A: Let's break down this problem: 1. How much older is Rena than Pam currently? 2. How old is Pam now? 1. Since Rena will be 5 years older than Pam in 10 years, she must be 5 years older than Pam now as well. The answer is 5.

2. If Pam is currently twice as young as Rena, that means that Rena is currently twice as old as Pam is. So if P stands for Pam's age now and R stands for Rena's age now, then we know that R = 2 * P And since Rena is 5 years older than Pam now, we know that R = P + 5. By substitution, we have P + 5 = 2 * P, which means that P = 5. The answer is 5. Therefore, the answer is (a).

Q: Cappuccinos cost $2, iced teas cost $3, cafe lattes cost $1.5 and espressos cost $1 each. Sandy orders some drinks for herself and some friends. She orders three cappuccinos, two iced teas, two cafe lattes, and two espressos. How much change does she receive back for a twenty-dollar bill? Answer Choices: (a) 20 (b) 17 (c) 3 (d) 1

A: Let's break down this problem: 1. How much did Sandy spend on drinks? 2. How much change does she receive back for a twenty-dollar bill?

1. Sandy ordered three cappuccinos, which cost $2 each, so she spent $2 * 3 = $6 on cappuccinos. She ordered two iced teas, which cost $3 each, so she spent $3 * 2 = $6 dollars on ice teas. She ordered two cafe lattes, which cost $1.5 each, so she spent $1.5 * 2 = $3 on cafe lattes. She ordered two espressos, which cost $1 each, so she spent $1 * 2 = $2 on espressos. So altogether, Sandy spent $6 + $6 + $3 + $2 = $17 on drinks. The answer is 17.

2. Sandy will get $20 - $17 = $3 as change. The answer is 3. Therefore, the answer is (c).

Q: question

A: Let's break down this problem:

**(2) The prompts of (L2M, $d_2$)**

**Four-shot exemplars for all the mathematical reasoning datasets excluding the AQuA dataset**:

Q: Four years ago, Kody was only half as old as Mohamed. If Mohamed is currently twice 30 years old, how old is Kody?

A: Let's break down this problem: 1. How old was Mohamed four years ago? 2. How old was Kody four years ago? 3. How old is Kody?

1. We were told that Mohamed is currently twice 30 years old, so he is currently 30 * 2 = 60 years old. That means that four years ago he must have been 60 - 4 = 56 years old. The answer is 56.

2. Four years ago, Kody was half as old as Mohamed, so Kody must have been 56 / 2 = 28 years old then. The answer is 28.

3. Since Kody was 28 years old four years ago, she must now be 28 + 4 = 32 years old. The answer is 32.

Q: Carla bought 2 bags of mini peanut butter cups on clearance. Each bag was $6.00 but was 75% off. How much did she spend on 2 bags of candy?

A: Let's break down this problem: 1. How much did she spend on 1 bag of candy? 2. How much did she spend on 2 bags of candy?

1. Each bag was $6.00 but was 75% off. So each bag cost $6.00 * (1 - 0.75) = $6.00 * 0.25 = $1.50.

2. Carla bought 2 bags. So she spent $1.50 * 2 = $3.00. The answer is 3.

Q: If Pam is currently twice as young as Rena is, and in 10 years Rena will be 5 years older than her, how old is Pam now?

A: Let's break down this problem: 1. How much older is Rena than Pam in 10 years? 2. How much older is Rena than Pam currently? 3. How old is Pam now?

1. We are told that Rena will be 5 years older than Pam in 10 years. The answer is 5.

2. So she must be 5 years older than Pam now as well. The answer is 5.

3. If Pam is currently twice as young as Rena, that means that Rena is currently twice as old as Pam is. So if P stands for Pam's age now and R stands for Rena's age now, then we know that R = 2 * P And since Rena is 5 years older than Pam now, we know that R = P + 5. By substitution, we have P + 5 = 2 * P, which means that P = 5. The answer is 5.

Q: Cappuccinos cost $2, iced teas cost $3, cafe lattes cost $1.5 and espressos cost $1 each. Sandy orders some drinks for herself and some friends. She orders three cappuccinos, two iced teas, two cafe lattes, and two espressos. How much change does she receive back for a twenty-dollar bill?

A: Let's break down this problem: 1. How much did the cappuccinos cost in total? 2. How much did the iced teas cost in total? 3. How much did the cafe lattes cost in total? 4. How much did the espressos cost in total? 5. How much did Sandy spend on drinks? 6. How much change does she receive back for a twenty-dollar bill?

1. Sandy ordered three cappuccinos, which cost $2 each, so she spent $2 * 3 = $6 on cappuccinos. The answer is 6.

2. She ordered two iced teas, which cost $3 each, so she spent $3 * 2 = $6 dollars on ice teas. The answer is 6.

3. She ordered two cafe lattes, which cost $1.5 each, so she spent $1.5 * 2 = $3 on cafe lattes. The answer is 3.

4. She ordered two espressos, which cost $1 each, so she spent $1 * 2 = $2 on espressos. The answer is 2.

5. So altogether, Sandy spent $6 + $6 + $3 + $2 = $17 on drinks. The answer is 17.

6. Sandy will get $20 - $17 = $3 as change. The answer is 3.

Q: {question}

A: Let's break down this problem:

**Four-shot exemplars for the AQuA dataset**:

Q: Four years ago, Kody was only half as old as Mohamed. If Mohamed is currently twice 30 years old, how old is Kody? Answer Choices: (a) 33 (b) 32 (c) 16 (d) 20

A: Let's break down this problem: 1. How old was Mohamed four years ago? 2. How old was Kody four years ago? 3. How old is Kody?

1. We were told that Mohamed is currently twice 30 years old, so he is currently 30 * 2 = 60 years old. That means that four years ago he must have been 60 - 4 = 56 years old. The answer is 56.

2. Four years ago, Kody was half as old as Mohamed, so Kody must have been 56 / 2 = 28 years old then. The answer is 28.

3. Since Kody was 28 years old four years ago, she must now be 28 + 4 = 32 years old. The answer is 32. Therefore, the answer is (b).

Q: Carla bought 2 bags of mini peanut butter cups on clearance. Each bag was $6.00 but was 75% off. How much did she spend on 2 bags of candy? Answer Choices: (a) 1.5 (b) 6 (c) 5 (d) 3

A: Let's break down this problem: 1. How much did she spend on 1 bag of candy? 2. How much did she spend on 2 bags of candy?

1. Each bag was $6.00 but was 75% off. So each bag cost $6.00 * (1 - 0.75) = $6.00 * 0.25 = $1.50.

2. Carla bought 2 bags. So she spent $1.50 * 2 = $3.00. The answer is 3. Therefore, the answer is (d).

Q: If Pam is currently twice as young as Rena is, and in 10 years Rena will be 5 years older than her, how old is Pam now? Answer Choices: (a) 5 (b) 10 (c) 15 (d) 20 (e) 30

A: Let's break down this problem: 1. How much older is Rena than Pam in 10 years? 2. How much older is Rena than Pam currently? 3. How old is Pam now?

1. We are told that Rena will be 5 years older than Pam in 10 years. The answer is 5.

2. So she must be 5 years older than Pam now as well. The answer is 5.

3. If Pam is currently twice as young as Rena, that means that Rena is currently twice as old as Pam is. So if P stands for Pam's age now and R stands for Rena's age now, then we know that R = 2 * P And since Rena is 5 years older than Pam now, we know that R = P + 5. By substitution, we have P + 5 = 2 * P, which means that P = 5. The answer is 5. Therefore, the answer is (a).

Q: Cappuccinos cost $2, iced teas cost $3, cafe lattes cost $1.5 and espressos cost $1 each. Sandy orders some drinks for herself and some friends. She orders three cappuccinos, two iced teas, two cafe lattes, and two espressos. How much change does she receive back for a twenty-dollar bill? Answer Choices: (a) 20 (b) 17 (c) 3 (d) 1

A: Let's break down this problem: 1. How much did the cappuccinos cost in total? 2. How much did the iced teas cost in total? 3. How much did the cafe lattes cost in total? 4. How much did the espressos cost in total? 5. How much did Sandy spend on drinks? 6. How much change does she receive back for a twenty-dollar bill?

1. Sandy ordered three cappuccinos, which cost $2 each, so she spent $2 * 3 = $6 on cappuccinos. The answer is 6.

2. She ordered two iced teas, which cost $3 each, so she spent $3 * 2 = $6 dollars on ice teas. The answer is 6.

3. She ordered two cafe lattes, which cost $1.5 each, so she spent $1.5 * 2 = $3 on cafe lattes. The answer is 3.

4. She ordered two espressos, which cost $1 each, so she spent $1 * 2 = $2 on espressos. The answer is 2.

5. So altogether, Sandy spent $6 + $6 + $3 + $2 = $17 on drinks. The answer is 17.

6. Sandy will get $20 - $17 = $3 as change. The answer is 3. Therefore, the answer is (c).

Q: question

A: Let's break down this problem:

**(3) The prompts of (L2M, $d_3$)**

**Four-shot exemplars for all the mathematical reasoning datasets excluding the AQuA dataset**:

Q: Four years ago, Kody was only half as old as Mohamed. If Mohamed is currently twice 30 years old, how old is Kody?

A: Let's break down this problem: 1. How old is Mohamed currently? 2. How old was Mohamed four years ago? 3. How old was Kody four years ago? 4. How old is Kody?

1. We were told that Mohamed is currently twice 30 years old, so he is currently 30 * 2 = 60 years old. The answer is 60.

2. That means that four years ago he must have been 60 - 4 = 56 years old. The answer is 56.

3. Four years ago, Kody was half as old as Mohamed, so Kody must have been 56 / 2 = 28 years old then. The answer is 28.

4. Since Kody was 28 years old four years ago, she must now be 28 + 4 = 32 years old. The answer is 32.

Q: Carla bought 2 bags of mini peanut butter cups on clearance. Each bag was $6.00 but was 75% off. How much did she spend on 2 bags of candy?

A: Let's break down this problem: 1. What was the original price of each bag? 2. What was the discount ratio of each bag? 3. How much did she spend on 1 bag of candy? 4. How much did she spend on 2 bags of candy?

1. Each bag was originally $6.00. The answer is 6.

2. But it was 75% off. The answer is 0.75.

3. So each bag cost $6.00 * (1 - 0.75) = $6.00 * 0.25 = $1.50. The answer is 1.50.

4. Carla bought 2 bags. So she spent $1.50 * 2 = $3.00. The answer is 3.

Q: If Pam is currently twice as young as Rena is, and in 10 years Rena will be 5 years older than her, how old is Pam now?

A: Let's break down this problem: 1. How much older is Rena than Pam in 10 years? 2. How much older is Rena than Pam currently? 3. Which equations need to be constructed to calculate Pam's age? 4. How old is Pam now?

1. We are told that Rena will be 5 years older than Pam in 10 years. The answer is 5.

2. So she must be 5 years older than Pam now as well. The answer is 5.

3. If Pam is currently twice as young as Rena, that means that Rena is currently twice as old as Pam is. So if P stands for Pam's age now and R stands for Rena's age now, then we know that R = 2 * P And since Rena is 5 years older than Pam now, we know that R = P + 5. The answer is R = 2 * P and R = P + 5.

4. By substitution, we have P + 5 = 2 * P, which means that P = 5. The answer is 5.

Q: Cappuccinos cost $2, iced teas cost $3, cafe lattes cost $1.5 and espressos cost $1 each. Sandy orders some drinks for herself and some friends. She orders three cappuccinos, two iced teas, two cafe lattes, and two espressos. How much change does she receive back for a twenty-dollar bill?

A: Let's break down this problem: 1. How many cappuccinos did Sandy order? 2. How much did the cappuccinos cost in total? 3. How many iced teas did Sandy order? 4. How much did the iced teas cost in total? 5. How many cafe lattes did Sandy order? 6. How much did the cafe lattes cost in total? 7. How many espressos did Sandy order? 8. How much did the espressos cost in total? 9. How much did Sandy spend on all drinks in total? 10. How much change does she receive back for a twenty-dollar bill?

1. Sandy ordered three cappuccinos. The answer is 3.

2. Each cappuccino cost $2 each, so she spent $2 * 3 = $6 on cappuccinos. The answer is 6.

3. She ordered two iced teas. The answer is 2.

4. Each iced tea cost $3 each, so she spent $3 * 2 = $6 dollars on ice teas. The answer is 6.

5. She ordered two cafe lattes. The answer is 2.

6. Each cafe latte cost $1.5 each, so she spent $1.5 * 2 = $3 on cafe lattes. The answer is 3.

7. She ordered two espressos. The answer is 2.

8. Each espressos cost $1 each, so she spent $1 * 2 = $2 on espressos. The answer is 2.

9. So altogether, Sandy spent $6 + $6 + $3 + $2 = $17 on drinks. The answer is 17.

10. Sandy will get $20 - $17 = $3 as change. The answer is 3.

Q: {question}

A: Let's break down this problem:

**Four-shot exemplars for the AQuA dataset**:

Q: Four years ago, Kody was only half as old as Mohamed. If Mohamed is currently twice 30 years old, how old is Kody? Answer Choices: (a) 33 (b) 32 (c) 16 (d) 20

A: Let's break down this problem: 1. How old is Mohamed currently? 2. How old was Mohamed four years ago? 3. How old was Kody four years ago? 4. How old is Kody?

1. We were told that Mohamed is currently twice 30 years old, so he is currently 30 * 2 = 60 years old. The answer is 60.

2. That means that four years ago he must have been 60 - 4 = 56 years old. The answer is 56.

3. Four years ago, Kody was half as old as Mohamed, so Kody must have been 56 / 2 = 28 years old then. The answer is 28.

4. Since Kody was 28 years old four years ago, she must now be 28 + 4 = 32 years old. The answer is 32. Therefore, the answer is (b).

Q: Carla bought 2 bags of mini peanut butter cups on clearance. Each bag was $6.00 but was 75% off. How much did she spend on 2 bags of candy? Answer Choices: (a) 1.5 (b) 6 (c) 5 (d) 3

A: Let's break down this problem: 1. What was the original price of each bag? 2. What was the discount ratio of each bag? 3. How much did she spend on 1 bag of candy? 4. How much did she spend on 2 bags of candy?

1. Each bag was originally $6.00. The answer is 6.

2. But it was 75% off. The answer is 0.75.

3. So each bag cost $6.00 * (1 - 0.75) = $6.00 * 0.25 = $1.50. The answer is 1.50.

4. Carla bought 2 bags. So she spent $1.50 * 2 = $3.00. The answer is 3. Therefore, the answer is (d).

Q: If Pam is currently twice as young as Rena is, and in 10 years Rena will be 5 years older than her, how old is Pam now? Answer Choices: (a) 5 (b) 10 (c) 15 (d) 20 (e) 30

A: Let's break down this problem: 1. How much older is Rena than Pam in 10 years? 2. How much older is Rena than Pam currently? 3. Which equations need to be constructed to calculate Pam's age? 4. How old is Pam now?

1. We are told that Rena will be 5 years older than Pam in 10 years. The answer is 5.

2. So she must be 5 years older than Pam now as well. The answer is 5.

3. If Pam is currently twice as young as Rena, that means that Rena is currently twice as old as Pam is. So if P stands for Pam's age now and R stands for Rena's age now, then we know that R = 2 * P And since Rena is 5 years older than Pam now, we know that R = P + 5. The answer is R = 2 * P and R = P + 5.

4. By substitution, we have P + 5 = 2 * P, which means that P = 5. The answer is 5. Therefore, the answer is (a).

Q: Cappuccinos cost $2, iced teas cost $3, cafe lattes cost $1.5 and espressos cost $1 each. Sandy orders some drinks for herself and some friends. She orders three cappuccinos, two iced teas, two cafe lattes, and two espressos. How much change does she receive back for a twenty-dollar bill? Answer Choices: (a) 20 (b) 17 (c) 3 (d) 1

A: Let's break down this problem: 1. How many cappuccinos did Sandy order? 2. How much did the cappuccinos cost in total? 3. How many iced teas did Sandy order? 4. How much did the iced teas cost in total? 5. How many cafe lattes did Sandy order? 6. How much did the cafe lattes cost in total? 7. How many espressos did Sandy order? 8. How much did the espressos cost in total? 9. How much did Sandy spend on all drinks in total? 10. How much change does she receive back for a twenty-dollar bill?

1. Sandy ordered three cappuccinos. The answer is 3.

2. Each cappuccino cost $2 each, so she spent $2 * 3 = $6 on cappuccinos. The answer is 6.

3. She ordered two iced teas. The answer is 2.

4. Each iced tea cost $3 each, so she spent $3 * 2 = $6 dollars on ice teas. The answer is 6.

5. She ordered two cafe lattes. The answer is 2.

6. Each cafe latte cost $1.5 each, so she spent $1.5 * 2 = $3 on cafe lattes. The answer is 3.

7. She ordered two espressos. The answer is 2.

8. Each espressos cost $1 each, so she spent $1 * 2 = $2 on espressos. The answer is 2.

9. So altogether, Sandy spent $6 + $6 + $3 + $2 = $17 on drinks. The answer is 17.

10. Sandy will get $20 - $17 = $3 as change. The answer is 3. Therefore, the answer is (c).

Q: question

A: Let's break down this problem:

## A.12.6   PROGRESSIVE-HINT PROMPTING (PHP)

**Four-shot exemplars for the mathematical reasoning datasets excluding the AQuA dataset**:

Q: Four years ago, Kody was only half as old as Mohamed. If Mohamed is currently twice 30 years old, how old is Kody? (Hint: The answer is near to 32).

A: We know the Answer Hints: 32. With the Answer Hints: 32, we will answer the question. We were told that Mohamed is currently twice 30 years old, so he is currently 30 * 2 = 60 years old. That means that four years ago he must have been 60 - 4 = 56 years old. Four years ago, Kody was half as old as Mohamed, so Kody must have been 56 / 2 = 28 years old then. Since Kody was 28 years old four years ago, she must now be 28 + 4 = 32 years old. The answer is 32.

Q: Carla bought 2 bags of mini peanut butter cups on clearance. Each bag was $6.00 but was 75% off. How much did she spend on 2 bags of candy? (Hint: The answer is near to 2, 5).

A: We know the Answer Hints: 2, 5. With the Answer Hints: 2, 5, we will answer the question. Each bag was $6.00 but was 75% off. So each bag cost $6.00 * (1 - 0.75) = $6.00 * 0.25 = $1.50. Carla bought 2 bags. So she spent $1.50 * 2 = $3.00. The answer is 3.

Q: If Pam is currently twice as young as Rena is, and in 10 years Rena will be 5 years older than her, how old is Pam now? (Hint: The answer is near to 10, 5).

A: We know the Answer Hints: 10, 5. With the Answer Hints: 10, 5, we will answer the question. Since Rena will be 5 years older than Pam in 10 years, she must be 5 years older than Pam now as well. If Pam is currently twice as young as Rena, that means that Rena is currently twice as old as Pam is. So if P stands for Pam's age now and R stands for Rena's age now, then we know that R = 2 * P And since Rena is 5 years older than Pam now, we know that R = P + 5. By substitution, we have P + 5 = 2 * P, which means that P = 5. The answer is 5.

Q: Cappuccinos cost $2, iced teas cost $3, cafe lattes cost $1.5 and espressos cost $1 each. Sandy orders some drinks for herself and some friends. She orders three cappuccinos, two iced teas, two cafe lattes, and two espressos. How much change does she receive back for a twenty-dollar bill? (Hint: The answer is near to 3).

A: We know the Answer Hints: 3. With the Answer Hints: 3, we will answer the question. Sandy ordered three cappuccinos, which cost $2 each, so she spent $2 * 3 = $6 on cappuccinos. She ordered two iced teas, which cost $3 each, so she spent $3 * 2 = $6 dollars on ice teas. She ordered two cafe lattes, which cost $1.5 each, so she spent $1.5 * 2 = $3 on cafe lattes. She ordered two espressos, which cost $1 each, so she spent $1 * 2 = $2 on espressos. So altogether, Sandy spent $6 + $6 + $3 + $2 = $17 on drinks, which means that sandy will get $20 - $17 = $3 as change. The answer is 3.

**Four-shot exemplars for the AQuA dataset**:

Q: Four years ago, Kody was only half as old as Mohamed. If Mohamed is currently twice 30 years old, how old is Kody? (Hint: The answer is near to (b)). Answer Choices: (a) 33 (b) 32 (c) 16 (d) 20

A: We know the Answer Hints: (b). With the Answer Hints: (b), we will answer the question. We were told that Mohamed is currently twice 30 years old, so he is currently 30 * 2 = 60 years old. That means that four years ago he must have been 60 - 4 = 56 years old. Four years ago, Kody was half as old as Mohamed, so Kody must have been 56 / 2 = 28 years old then. Since Kody was 28 years old four years ago, she must now be 28 + 4 = 32 years old. The answer is (b).

Q: Carla bought 2 bags of mini peanut butter cups on clearance. Each bag was $6.00 but was 75% off. How much did she spend on 2 bags of candy? (Hint: The answer is near to (a), (c)). Answer Choices: (a) 1.5 (b) 6 (c) 5 (d) 3

A: We know the Answer Hints: (a), (c). With the Answer Hints: (a), (c), we will answer the question. Each bag was $6.00 but was 75% off. So each bag cost $6.00 * (1 - 0.75) = $6.00 * 0.25 = $1.50. Carla bought 2 bags. So she spent $1.50 * 2 = $3.00. The answer is (d).

Q: If Pam is currently twice as young as Rena is, and in 10 years Rena will be 5 years older than her, how old is Pam now? (Hint: The answer is near to (a), (b)). Answer Choices: (a) 5 (b) 10 (c) 15 (d) 20 (e) 30

A: We know the Answer Hints: (a), (b). With the Answer Hints: (a), (b), we will answer the question. Since Rena will be 5 years older than Pam in 10 years, she must be 5 years older than Pam now as well. If Pam is currently twice as young as Rena, that means that Rena is currently twice as old as Pam is. So if P stands for Pam's age now and R stands for Rena's age now, then we know that R = 2 * P And since Rena is 5 years older than Pam now, we know that R = P + 5. By substitution, we have P + 5 = 2 * P, which means that P = 5. The answer is (a).

Q: Cappuccinos cost $2, iced teas cost $3, cafe lattes cost $1.5 and espressos cost $1 each. Sandy orders some drinks for herself and some friends. She orders three cappuccinos, two iced teas, two cafe lattes, and two espressos. How much change does she receive back for a twenty-dollar bill? (Hint: The answer is near to (c)). Answer Choices: (a) 20 (b) 17 (c) 3 (d) 1

A: We know the Answer Hints: (c). With the Answer Hints: (c), we will answer the question. Sandy ordered three cappuccinos, which cost $2 each, so she spent $2 * 3 = $6 on cappuccinos. She ordered two iced teas, which cost $3 each, so she spent $3 * 2 = $6 dollars on ice teas. She ordered two cafe lattes, which cost $1.5 each, so she spent $1.5 * 2 = $3 on cafe lattes. She ordered two espressos, which cost $1 each, so she spent $1 * 2 = $2 on espressos. So altogether, Sandy spent $6 + $6 + $3 + $2 = $17 on drinks, which means that sandy will get $20 - $17 = $3 as change. The answer is (c).

