# OpenReview forum: "Adaptive-Solver Framework for Dynamic Strategy Selection in Large Language Model Reasoning"
_ICLR.cc/2024/Conference — ICLR 2024 Conference Withdrawn Submission_

### Official Review · Reviewer_63XR · 2023-10-30

**Soundness:** 2 fair
**Presentation:** 2 fair
**Contribution:** 2 fair
**Rating:** 3
**Confidence:** 3

**Summary:**

The paper proposes an adaptive approach to using LLMs that may switch between
different LLMs, change prompting, and decompose the problem based on the
observed performance of initial or partial solutions. The authors describe their
framework and evaluate it empirically.

**Strengths:**

The paper is well written and explores an interesting idea.

**Weaknesses:**

Details of it are unclear. In particular the decomposition seems to rely on
manually defined granularities, i.e. a given problem cannot be decomposed
automatically. This imposes a significant burden on the user. The results of the
empirical evaluation seem to suggest that this decomposition is often crucial to
achieving good performance; this should be discussed in more detail.

**Questions:**

As a user, how would I decide how to decompose, how many levels are needed, and what the
levels represent?

---

> ### Author Response · Authors · 2023-11-21
> **Response to Reviewer 63XR (1/2)**
>
> Thank you for your detailed questions. We will respond to your questions as follows:
>
> **W1: About additional workload on users due to manually defined granularities.**
>
> **A: Compared to L2M, our method does not place significantly greater burdens on users**. The implementation of decomposition granularity adaptation involves two steps: 1) Crafting multiple variations of the L2M [1] prompting method to guide an LLM in decomposing a problem at different levels of granularity. 2) Initiating the LLM from a coarser level of decomposition and systematically progressing towards finer levels ultimately helps determine the most suitable granularity.
>
> Once the candidate decomposition granularities are established in the first step, the optimal decomposition granularity for different problems is dynamically determined in the second step. While the first step requires "manually defined granularities", it primarily entails making minor adjustments to the prompts of L2M. These adjustments can be systematically approached, as we will explain shortly. Consequently, in comparison to L2M, our method does not place significantly greater burdens on users.
>
> [1] Zhou D, Schärli N, Hou L, et al. Least-to-most prompting enables complex reasoning in large language models. ICLR 2023.
>
> **Now we introduce how to construct prompt variants of L2M.** To illustrate, consider the following example question: *Cappuccinos cost 2 dollars, iced teas cost 3 dollars, cafe lattes cost 1.5 dollars and espressos cost 1 dollar each. Sandy orders some drinks for herself and some friends. She orders three cappuccinos, two iced teas, two cafe lattes, and two espressos. How much change does she receive back for a twenty-dollar bill?*
>
> L2M does not control the decomposition granularity deliberately and its decomposition for the example question is as follows: *1. How much did the cappuccinos cost in total? 2. How much did the iced teas cost in total? 3. How much did the cafe lattes cost in total? 4. How much did the espressos cost in total? 5. How much did Sandy spend on drinks? 6. How much change does she receive back for a twenty-dollar bill?*
>
> To construct L2M's variants, we first decompose the question hierarchically, as shown in **Figure 5 (Appendix A.11)** in the revised paper. And then we obtain the decompositions of L2M's variants with the following steps:
>
> (1) First, we extract the problem and sub-problems from the first layer of decomposition. Then, serialize them from bottom to top to obtain the sequence of sub-problems in (L2M, $d_1$)'s prompt: *1. How much did Sandy spend on drinks? 2. How much change does she receive back for a twenty-dollar bill?* **(2 sub-questions)**
>
> (2) Similarly, we extract the problem and sub-problems from the first two layers of decomposition and then serialize them to obtain the sequence of sub-problems in (L2M, $d_2$)'s prompt: *1. How much did the cappuccinos cost in total? 2. How much did the iced teas cost in total? 3. How much did the cafe lattes cost in total? 4. How much did the espressos cost in total? 5. How much did Sandy spend on drinks? 6. How much change does she receive back for a twenty-dollar bill?* **(6 sub-questions)**
>
> (3) Likewise, we extract the problem and sub-problems from the three layers of decomposition and serialize them to obtain the sequence of sub-problems in (L2M, $d_3$)'s prompt: *1. How many cappuccinos did Sandy order? 2. How much did the cappuccinos cost in total? 3. How many iced teas did Sandy order? 4. How much did the iced teas cost in total? 5. How many cafe lattes did Sandy order? 6. How much did the cafe lattes cost in total? 7. How many espressos did Sandy order? 8. How much did the espressos cost in total? 9. How much did Sandy spend on all drinks in total? 10. How much change does she receive back for a twenty-dollar bill?* **(10 sub-questions)**
>
> We have also added the above discussion into the Appendix A.11 of the revised version of this paper.

---

> > ### Author Response · Authors · 2023-11-21
> > **Response to Reviewer 63XR (2/2)**
> >
> > **W2: Discussion about the importance of decomposition to good performance**
> >
> > **A: We will discuss about the reason why decomposition is crucial to performance and the necessity of our method.**
> > The results in Table 3 indeed suggest that decomposition granularity in in-context examples affects the performance. The reason is that a decomposition that is too coarse may oversimplify the main question, while an excessively detailed breakdown can increase the risk of decomposition errors. Consequently, inappropriate decomposition granularity may lead to a decrease in performance. Besides, fixed-granularity decomposition methods exhibit inconsistent performance across various datasets. This inconsistency arises from the fact that problems with differing complexities necessitate distinct levels of decomposition granularity. Consequently, fixed-granularity decomposition methods lack the ability to adapt to these diverse problem requirements. Hence, we introduce an adaptive-granularity method, and the results presented in Table 3 validate its effectiveness.
> >
> > ---
> >
> > **Q1: Question about how to decompose, how many levels are needed and what the levels represent.**
> >
> > We will answer the three questions one by one.
> >
> > (1) For users, they need to manually decompose a few example questions when constructing prompts. The decomposition procedure has been illustrated in answering W1 earlier. In summary, the users need to decompose an example question hierarchically, and then extract and serialize sub-problems.
> >
> > (2) The question of "how many levels are needed" may lack a definitive gold standard. Empirically, we found that three levels of decomposition prove to be a suitable choice, as demonstrated in our experiments. This number strikes a balance, being sufficiently fine to significantly reduce the complexity of the main question, without becoming overly detailed and risking decomposition errors.
> >
> > (3) The term "the levels" represent the levels of detail or granularity of decomposition. Referring back to Figure 5 (Appendix A.11), the one level of decomposition corresponds to (L2M, $d_1$)'s decomposition and the three levels of decomposition corresponds to (L2M, $d_3$)'s decomposition.

---

> > > ### Comment · Reviewer_63XR · 2023-11-21
> > >
> > > Thank you for the clarifications. It still seems that this is crucial to achieve good performance and relies on the user, with little guidance on how to do it.

---

> > > > ### Author Response · Authors · 2023-11-22
> > > > **Response to Reviewer 63XR**
> > > >
> > > > Thanks for your comments! Model creator needs to construct prompts according to our guidelines, for users, no additional actions are required compared to other methods such as querying ChatGPT directly.
> > > >
> > > > 1\. **Only the example questions in prompts need manual decomposition**. Is your concern about increased user burden due to the belief that our method necessitates manual problem decomposition for all questions, including example questions and test questions? In fact, our method only requires manual decomposition for a small set of example questions, such as 4 example questions used in our paper. **For all test questions, our method can decompose the problems and choose the appropriate decomposition granularity automatically.**
> > > >
> > > > 2\. **Guidelines for Constructing Prompts**. Using a small set of example questions, various prompts are constructed to guide an LLM in decomposing problems at different levels of granularity. The guidelines for constructing these prompts are as follows:
> > > >
> > > >    1\) Perform hierarchical problem decomposition (assuming three levels) as illustrated in **Figure 5 (Appendix A.11)**.
> > > >
> > > >    2\) Construct the first prompt by selecting sub-problems from the initial level of decomposition, in addition to the main problem.
> > > >
> > > >    3\) Construct the second prompt by selecting sub-problems from the first two levels of decomposition, in addition to the main problem.
> > > >
> > > >    4\) Construct the third prompt by selecting sub-problems from all three levels of decomposition, in addition to the main problem.
> > > >
> > > >    For more details about extracting sub-problems for different prompts, please refer to **Appendix A.11** of the revised paper.
> > > >
> > > > 3\. **How to decompose problems automatically**. For all test questions, to achieve automatic problem decomposition and granularity selection, our method automatically performs the following steps:
> > > >
> > > >    1\) Use the first prompt mentioned above to guide the LLM to decompose a problem at a coarse granularity and then solve it.
> > > >
> > > >    2\) If the solution generated from step (1) is evaluated as incorrect, use the second prompt to guide the LLM to decompose the problem at a moderate granularity and solve it. Otherwise, conclude the solving process.
> > > >
> > > >    3\) If the solution generated from step (2) is also evaluated as incorrect, use the third prompt to guide the LLM to decompose the problem at a fine granularity and solve it. Otherwise, conclude the solving process.
> > > >
> > > >    4\) If all the solutions from the three steps fail to meet the evaluation criteria, use the solution with the highest evaluation score as the final solution.
> > > >
> > > >    In this manner, our method can determine the appropriate decomposition granularity for each specific problem automatically.
> > > >
> > > >    For more details about the three prompts, please refer to **Appendix A.12.5** of the revised paper.

---

> > > > > ### Comment · Reviewer_63XR · 2023-11-22
> > > > >
> > > > > I'm concerned about the training decompositions. It sounds like if the user provides the "wrong" decompositions, performance will be bad.

---

> > > > > > ### Author Response · Authors · 2023-11-23
> > > > > >
> > > > > > Thanks for your response !
> > > > > >
> > > > > > Could we interpret the term "'wrong' decomposition" as "the decomposition that performs bad on certain dataset" or "the inappropriate decomposition granularity for certain dataset" ?
> > > > > > For example, in Table 3, L2 performs bad on the GSM8K dataset, so can we consider L2 to be the "wrong" decomposition for the dataset GSM8K?
> > > > > >
> > > > > > If our interpretation aligns with your intended meaning, we hope the following explanation addresses your concern:
> > > > > >
> > > > > > 1\) Upon examining Table 3, we observe that for methods utilizing fixed-granularity decomposition, the selection of an appropriate decomposition is crucial. This is because these methods may excel on some datasets but perform poorly on others. If the user provides inappropriate decomposition, performance will be bad.
> > > > > >
> > > > > > 2\) In contrast, our method integrates different decompositions with varying levels of granularity, eliminating the necessity for users to manually choose an appropriate decomposition or decomposition granularity.

---

> ### Author Response · Authors · 2023-11-23
>
> Dear Reviewer 63XR,
>
> We sincerely appreciate the time and effort you dedicated to reviewing our paper and providing valuable feedback through multiple rounds.
>
> We would like to inquire whether our responses have addressed your concern about the "wrong" decomposition. We are open to further discussion and would be delighted to address any additional feedback you may have.

---

### Official Review · Reviewer_5Pco · 2023-10-31

**Soundness:** 2 fair
**Presentation:** 4 excellent
**Contribution:** 2 fair
**Rating:** 3
**Confidence:** 4

**Summary:**

This paper introduces an adaptive LLM solver method that iterates through different methods to arrive at a solution. Adaptive solver first checks if a given solution is accurate before trying different adaptations.  The adaptations are model, prompting and decomposition granularity. Experiments show how the different adaptations affect performance on various reasoning datasets.

**Strengths:**

- Originality: The idea of the framework introduced is unique for its flexibility. Most papers focus on trying to improve one of the given adaptations while the adaptive solver takes a different approach.
- Significance: Searching or solving for the best way to approach a particular question is an important line of work, especially given the high cost of API and variety of inputs.
- The paper is well presented. In particular, the conclusions from the analysis and experiments are easy to find.

**Weaknesses:**

- There are a limited number of options for the solver and the method requires a lot of pre-processing (to write the in-context examples) for the prompting method.
- The implementation is not as novel as the original idea. To the best of my understanding, the solver goes through different methods and chooses the best one. A significant improvement would come from making the solver more dynamic and based on the solution. Works that combine planning and LLMs are quite relevant.*
- The paper mentions that the number of solving rounds does not increase much but there is no discussion of the increase in inference time. Trying different approaches until a certain number of iterations has passed or some metric is satisfied will increase the inference time significantly. This could be problematic for real-time applications.


*Relevant papers:
- Wang, Lei, et al. "Plan-and-solve prompting: Improving zero-shot chain-of-thought reasoning by large language models." arXiv preprint arXiv:2305.04091 (2023).
- Hao, Shibo, et al. "Reasoning with language model is planning with world model." arXiv preprint arXiv:2305.14992 (2023).

**Questions:**

- For the Decomposition Granularity Adaptation experiment, what model is used? GPT-3.5? Is there a way to compare this with GPT-4?
- How is Decomposition Granularity different from prompting? From Figure 1, c and d look quite similar.
- Were there experiments using a larger solver list? From Table 1, each of the 3 solvers has the best performance in at least one of the datasets.
- Results clearly depend on what is in the solver. Is there a way to choose what methods to put into a given solver?
- How did inference time change per types of adaptations?
- How were in-context examples chosen?

---

> ### Author Response · Authors · 2023-11-21
> **Response to Reviewer 5Pco (1/2)**
>
> We appreciate your detailed review and constructive feedback. We respond to your questions as follows:
>
> **W1: There are a limited number of options for the solver.**
>
> The Adaptive-Solver framework possesses strong adaptability and flexibility, allowing for the inclusion of any number of solvers in its solver list. It is easy to obtain a different solver by modifying components such as the LLM model, prompting method, or specific parameters (i.e., the sampling quantity in the self-consistency method) within a solver. Expansion is straightforward, requiring only the addition of a solver to the solver list. In our research, we have explored three LLM models (GPT3.5, GPT4, GLM2) and four prompting methods (CoT, L2M, ZeroCoT, PS). Furthermore, the framework can incorporate other LLM models (e.g., Claude2, Llamma2) and prompting methods (e.g., PoT, ToT) to construct a diverse set of solvers.
>
> **W2: A lot of pre-processing required to write the in-context examples**
>
> Our Adaptive-Solver framework demonstrates adaptability to various types of prompting method, whether zero-shot or few-shot. In both prompting method adaptation and model adaptation, we explored zero-shot prompting methods, including zero-shot CoT and PS, which do not need in-context examples. For decomposition granularity adaptation, our approach is based on L2M that is a few-shot method. Our method's prompts can be obtained with minor adjustments to L2M's prompts, without requiring a significantly greater workload. A systematic approach detailing how to derive our prompts from L2M is provided in Appendix A.11.
>
> **W3: About the novelty of our implementation and a more dynamic solver**
>
> We believe that adapting the solver in a more dynamic and solution-oriented manner holds promise for improvement. To the best of our knowledge, the Adaptive-Solver framework is novel and has not been introduced in the existing literature of LLM. We adopt a straightforward adaptation strategy that is both simple and effective. The exploration of a more dynamic adaptation strategy within this framework presents a promising avenue for future research endeavors.
>
> **W4/Q5: About inference time change of our method**
>
> According to your insightful suggestion, we discuss about the increase in inference time below. We have documented the average inference time per problem associated with various prompting methods, as presented in both **Table 9 and Table 10 (Appendix A.4)** of the revised paper. We observe that, akin to the number of solving rounds, there is not a significant increase in inference time. Specifically, the average inference time of AS-P (AS-D and AS-PD) is no more than 1.6 times that of COT* (L2M*). The average inference time of AS-M is no more than 1.2 times that of GPT4. The rationale behind this is that the majority of problems are resolved by the initial solver, with the subsequent solvers only being invoked in a few necessary cases.
> We have added this discussion into the Appendix A.4 of the revised version of this paper.
>
> ---
>
> **Q1: The LLM model for the Decomposition Granularity Adaptation experiment and comparison with GPT-4**
>
> We clarify the model information and do the comparison with GPT-4.
>
> (1) **Model information**. In the ablation experiment of Decomposition Granularity Adaptation, we utilized GPT-3.5-turbo-0301, as detailed in Appendix A.2 "Implementation Details". We have also added the information into section 4.1 "experiment setup" before this experiment.
>
> (2) **Comparison with GPT-4**. We have conducted the experiment with GPT-4 on three datasets, and the comparison with GPT-3.5 is demonstrated in the following table. On GPT4, our method still achieves the best performance in terms of average results. Besides, our method demonstrates a more stable performance. In contrast to fixed-granularity methods, exemplified by [L1, L1, L1], which excels on GSM8K but underperform on SVAMP, our method maintains a more consistent level of performance across different datasets.
>
> | | | GSM8K | SVAMP | AQuA | Average|
> | :---:| :---:| :---:| :---:| :---:| :---:|
> | GPT3.5 | [L, L, L] | 86.1 | 87 | 61.9 | 78.3 |
> | | [L1, L1, L1] | 87 | 86.1 | 62 | 78.4 |
> | | [L2, L2, L2] | 85.2 | 85 | 62.8 | 77.7 |
> | | [L3, L3, L3] | 85.5 | 88.3 | 62.4 | 78.7 |
> | | [L1, L2, L3] | 87.5 | 89 | 63.3 | 79.9 |
> GPT4 | [L, L, L] | 95.1 | 87.4 | 74.8 | 85.8 |
> | | [L1, L1, L1] | 96.3 | 87.4 | 76 | 86.6 |
> | | [L2, L2, L2] | 96.2 | 88.9 | 76.4 | 87.2 |
> | | [L3, L3, L3] | 95.6 | 92.8 | 76 | 88.1 |
> | | [L1, L2, L3] | 96.1 | 91.5 | 76.8 | 88.1 |

---

> > ### Comment · Reviewer_5Pco · 2023-11-23
> >
> > Thank you for the detailed explanations and additional experiments. After much consideration, I will increase my score from 3 to 5. I think this work has lots of potential but the empirical results are somewhat weak. In particular, for GPT-4, the [L3,L3,L3] prompting method has the same average performance as [L1,L2,L3].

---

> > > ### Author Response · Authors · 2023-11-23
> > >
> > > Dear Reviewer 5Pco,
> > >
> > > We greatly appreciate your time in reviewing our paper and are delighted with your decision to raise the score. We will improve the paper further in accordance with your suggestions.

---

> ### Author Response · Authors · 2023-11-21
> **Response to Reviewer 5Pco (2/2)**
>
> **Q2: How is Decomposition Granularity different from prompting? From Figure 1, c and d look quite similar.**
>
> We clarify the difference of Decomposition Granularity in different promptings. According to our understanding, the question pertains to the difference in decomposition granularity between the second prompt in (c) and the prompts in (d). In Figure 1 (c), the second prompting method employs a decomposition-based approach, specifically L2M. In Figure 1 (d), we utilize three L2M's variants as prompting methods, denoted as (L2M, $d_1$), (L2M, $d_2$), and (L2M, $d_3$), each representing different levels of decomposition granularity.
>
> In (c), we utilize the original L2M and incorporate identical in-context examples as presented in [1]. The original L2M lacks control over the decomposition granularity in its in-context examples, resulting in a mixture of decompositions at various granularities. In contrast, (L2M, $d_1$), (L2M, $d_2$), and (L2M, $d_3$) control the granularity at their decompositions. For the specific in-context examples of L2M and L2M's variants, please refer to Appendix A.12.4 and A.12.5.
>
> **Q3: Were there experiments using a larger solver list? From Table 1, each of the 3 solvers has the best performance in at least one of the datasets.**
>
> (1) **The meaning of the three solvers in Table 1**. AS-PD employed a solver list consisting of four solvers, namely [COT*, (L2M*, d1), (L2M*, d2), (L2M*, d3)]. In our experiments, we did not explore a solver list larger than this. AS-P represents prompting method adaptation, utilizing the solver list [COT*, L2M*]. AS-D represents decomposition granularity adaptation, employing the solver list [(L2M*, d1), (L2M*, d2), (L2M*, d3)].
>
> (2) **Explanation about the performance**. AS-D performs the best on the SVAMP and AddSub datasets, while AS-PD performs the best on the other datasets. AS-P performs equally well as AS-PD on the MultiArith dataset. This is attributed to AS-D commencing with (L2M*, d1), while AS-P and AS-PD begin with CoT*. As shown in Table 1, it is apparent that L2M (homogeneous to (L2M, d1)) outperforms CoT on the SVAMP and AddSub datasets. Consequently, AS-D, starting with (L2M*, d1), also exhibits the best performance on these two datasets. This implies that the effectiveness of the initial solver has a noteworthy impact on the ultimate performance. It is evident that our prompting method adaptation (based on CoT) and decomposition granularity method (based on L2M) notably improve CoT and L2M respectively. Furthermore, the combined approach (i.e., AS-PD), leveraging both prompting method adaptation and decomposition granularity adaptation, demonstrates the best overall performance.
>
> **Q4: Results clearly depend on what is in the solver. Is there a way to choose what methods to put into a given solver?**
>
> We provide two approaches for selecting methods to include in a given solver:
>
> (1) The first method is a comprehensive approach suitable for all types of adaptation. It involves sampling a small subset of the dataset as a validation set and experimenting with various permutations and combinations of solvers. This helps determine which solvers to include and the optimal order in the solver list.
>
> (2) The second method specifically addresses Model Adaptation, aiming to maintain performance while minimizing costs. Typically, as the model's capabilities increase, so does the associated cost. Therefore, a practical strategy involves constructing the solver list by arranging solvers from weaker to stronger based on the model's strength.
> We have added this discussion into Appendix A.7 of the revised version of this paper.
>
> **Q6: How were in-context examples chosen?**
>
> (1) In mathematical reasoning experiments, all few-shot prompting methods utilize the same set of four questions to generate in-context examples. These questions are derived from examples used in the problem decomposition of L2M [1].
>
> (2) In experiments involving symbolic reasoning and commonsense reasoning, all few-shot prompting methods employ the same four questions to create prompts. These questions are drawn from examples used in Plan-and-Solve [2].
>
> ---
>
> [1] Zhou D, Schärli N, Hou L, et al. Least-to-most prompting enables complex reasoning in large language models. ICLR 2023.
>
> [2] Wang L, Xu W, Lan Y, et al. Plan-and-Solve Prompting: Improving Zero-Shot Chain-of-Thought Reasoning by Large Language Models. ACL2023.

---

### Official Review · Reviewer_uCwU · 2023-10-31

**Soundness:** 2 fair
**Presentation:** 2 fair
**Contribution:** 2 fair
**Rating:** 5
**Confidence:** 4

**Summary:**

The paper presents an approach for combining multiple different strategies in order to solve problems using LLMs. This is akin to portfolio selection since there is generally no "one-size-fits-all" approach to solving problems. In order to do so, the authors propose the adaptive-solver (AS) framework for LLMs.

AS consists of three different adaptation strategies, (a) Model adaptation where the LLM models are changed from cheaper to more advanced albeit expensive models, (b) Prompting method adaptation wherein different prompting methods are utilized for problems, and finally decomposition granularity adaptation that tailors the decomposition granularity of prompts from coarse to finer.

An adaptation module consists of a portfolio of such solvers and the authors use an evaluation module with a consistency metric to determine the evaluation criteria for switching solvers. The authors then provide an empirical evaluation of their approach and perform several ablations of each of the modules.

**Strengths:**

1) The paper is generally well-written (even though the empirical could have been organized a bit better) and the ideas are expressed clearly

2) The idea of having a portfolio of such selection strategies makes sense since empirically it is known that there is usually not a single approach that can outperform others

**Weaknesses:**

The paper is quite interesting but it seems that the empirical evaluation section is (a) bit hard to follow, and more importantly (b) the results only show a marginal improvement over baselines.

a) The paper only improves over baselines by a nominal ~3% (Table 1). This does not seem very significant to me and is further exacerbated by the fact that different, hand-coded variations of AS are needed to outperform the baselines as such.

b) The paper claims that AS can cut down on API costs but a cost analysis vs baselines is not provided. Table 2 only provides cost analysis vs using two versions of GPT but does not include overall costs for the entire pipeline.

c) Similarly, Table 3 only shows marginal improvements for the decomposition granularity ablation.

Overall, the ablations are interesting but the process seems overly hand-coded with not enough improvements over the baselines. (Even the strategies for choosing solvers is driven by expert-knowledge). For example, how many times was strategy 1 (choose the last solver in the list) selected in your evaluation. Such information is missing in the main paper.

**Questions:**

Id like to thank the authors for their extensive experiments. I've listed my questions below. I hope that the authors can resolve my queries.

1. Could you please comment on (b) and provide a reason as to why overall costs for the entire pipeline are not included in the paper.

2. Currently, it feels like most of the experiments are ablations. I would have preferred to have seen results with a general AS solver list and a more comprehensive comparison with baselines.

3. I can understand the reason for the ablations but is there any reason as to why all baselines were not tried on for all datasets? For example, Table 2 only uses ZeroCoT for prompting and only the model adaptation is explained. I think that the overall efficacy of the pipeline can only be clearly determined when the pipeline is used everywhere and not selectively applied to different datasets. I appreciate the authors trying to reduce the # of variables but this only made the evaluation more confusing for me.

---

> ### Author Response · Authors · 2023-11-21
> **Response to Reviewer uCwU (1/2)**
>
> Thank you for your detailed review and constructive feedback. We respond to your questions as follows:
>
> **W1: The improvement is nominal over baselines in Table 1; different, hand-coded variations of AS are needed to outperform the baselines**
>
> A: We would like to talk more about the improvement and highlight advantages of our method.
>
> (1) **AS-PD alone consistently surpasses all the baselines**. Table 1 illustrates that all three of our adaptive methods outperform the baselines in terms of average performance. Specifically, AS-P, AS-D and AS-PD lead the best average performance (i.e., 85.8% of COT_SC) among baselines by 1.4%, 2% and 2.8%, respectively. AS-PD, in particular, surpasses the baselines across all datasets. We don't need different variations of AS to just outperform baselines. AS-PD alone consistently outperforms all baselines across all datasets. Particularly, on each dataset (from GSM8K to MultiArith), AS-PD leads the best result among baselines by 3.1%, 5.4%, 2.9%, 1.6%, 0.6%, 0%. Our approaches showcase superior overall performance and greater stability compared to the baselines. These findings underscore the practical utility of our approach in real-world scenarios.
>
> (2) **The improvements on commonsense reasoning and symbolic reasoning task are pronounced**. In the experiments of commonsense reasoning (CS dataset) and symbolic reasoning (LLC dataset), our method AS-P adopts zero-shot setting (refer to Table 7 in Appendix A.3). When compared to the zero-shot baselines (i.e., ZeroCoT and PS), AS-P shows significant improvements. Specifically, AS-P achieves an average enhancement of 10.4%, with an impressive 16.2% advantage specifically on the LLC dataset. Furthermore, even in comparison to few-shot baselines (i.e., CoT and CoT_SC), our method maintains a 3.8% advantage. These results highlight the notable effectiveness of our approach in these reasoning tasks.
>
> (3) **Significant reductions in cost**. Our approach is not only applicable for performance improvement but also for cost reduction. Experimental results in Table2 indicates that the model adaptation approach significantly reduces API costs (up to 50%) while maintaining superior performance.
>
> (4) **Framework extensibility**. Our work focuses more on the design of the Adaptive-Solver framework and validating its effectiveness through preliminary instantiation. This framework extends beyond the scope of the implementation in this paper. By incorporating more advanced or complementary prompting methods (e.g., PoT, ToT) into this framework, there is the potential to achieve higher performance. In an additional effort, we extend AS-PD (with the solver list as [CoT*, (L2M*, d1), (L2M*, d2), (L2M*, d3)]) to [CoT*, PS, (L2M*, d1), (L2M*, d2), (L2M*, d3)], by just adding PS into it. This modified implementation demonstrates an improved accuracy of 89.1%, surpassing the original accuracy of 88.6% on the GSM8K dataset. This suggests the possibility of further enhancing our method.
>
> **W2/Q1: About cost analysis vs baselines and overall costs for the entire pipeline**
>
> We would like to clarify what are the baselines for model adaptation experiment and the meaning of "Cost ($)" in Table 2.
>
> (1) **Cost analysis vs baselines was provided**. In model adaptation, we combine a weaker LLM model and a stronger LLM model to cut down the overall API cost while maintaining superior performance. Therefore, in this experiment the baselines are methods that using solely the stronger LLM model and methods using only the weaker LLM model. As shown in Table 2, our baselines are GPT4, GPT3.5* and [GPT3.5*, GPT3.5*], while our method is [GPT3.5*, GPT4]. The paper included a detailed cost analysis with these baselines in section 4.3.
> To enhance clarity, we have clarified the baselines of each type of adaptation in section 4.1 "experiment setup" of the revised version of this paper.
>
> (2) **Overall costs for the entire pipeline were included**. According to our understanding, "overall costs for the entire pipeline" in your comments refers to total costs of all solvers within one method. Take [GPT3.5*, GPT4] as an example, it refers to the total cost of invoking both GPT3.5* and GPT4. In Table 2, the entries under the "Cost ($)" column represent the overall costs for the entire pipeline, encompassing both the costs of invoking GPT3.5 and GPT4. Please let us know if we misunderstood your question.

---

> ### Author Response · Authors · 2023-11-21
> **Response to Reviewer uCwU (2/2)**
>
> **W3: Improvements for the decomposition granularity ablation are marginal.**
>
> We would like to clarify what the "improvement" in this experiment represents and highlight the advantage of our method.
>
> (1) **Clarification about the improvement**. To avoid being misunderstood as an improvement of the entire decomposition granularity adaptation method, we would like to clarify that: The entire decomposition granularity adaptation method is an enhancement based on L2M. If for evaluating the effectiveness of the entire method, a more reasonable comparison would be with L2M. The results in Table 1 show that the decomposition granularity method significantly outperform L2M (~10% improvement on GSM8K). This ablation experiment here aims to explore the performance improvement brought by gradually refining the decomposition granularity (just one feature of our method), compared to maintaining a constant granularity.
>
> (2) **Our method is more stable**. The results suggest that gradually refining the decomposition granularity yields better results than maintaining a constant granularity, although the improvement is not highly pronounced. Moreover, the results indicate that this feature contributes to enhancing the overall stability of the method. In the following table, the performance rankings of each method on each dataset are annotated (i.e., the integer within the parentheses). It is observed that methods with adaptive adjustment of decomposition granularity perform best on almost all datasets. However, other methods maintaining a constant granularity show greater fluctuations in rankings across different datasets. The adaptive adjustment of decomposition granularity integrates different granularities, relieving users from the burden of choosing a specific granularity. Furthermore, its performance stability suggests that it is more adaptable to different data distributions, providing an advantage when dealing with unknown data distributions in practical applications.
>
> | Method | GSM8K | SVAMP | AQuA | AddSub | SingleEq | MultiArith | Average|
> | :---:| :---:| :---:| :---:| :---:|:---:| :---:| :---:|
> [L, L, L] | 86.1 (3) | 87.0 (3) | 61.9 (5) | 92.4 (2) | 95.3 (3) | 96.3 (3) | 86.5 |
> [L1, L1, L1] | 87.0 (2) | 86.1 (4) | 62.0 (4) | 91.9 (4) | 95.6 (1) | 98.7 (1) | 86.9 |
> [L2, L2, L2] | 85.2 (4) | 85.0 (5) | 62.8 (2) | 89.9 (5) | 94.3 (4) | 95.2 (5) | 85.4 |
> [L3, L3, L3] | 85.5 (5) | 88.3 (2) | 62.4 (3) | 92.2 (3) | 95.4 (2) | 95.7 (4) | 86.6 |
> [L1, L2, L3] | 87.5 (1) | 89.0 (1) | 63.3 (1) | 92.9 (1) | 95.6 (1) | 98.4 (2) | 87.8 |
>
> **W4: how many times was the selection of strategy 1**
>
> In section 3.2 of our paper, we present two strategies for choosing the final solver in case none of the solvers meet the criteria. Strategy 1 is applied for model adaptation and prompting method adaptation, while strategy 2 is employed for decomposition granularity adaptation. The question "how many times was strategy 1 selected" is not entirely clear to us, as strategy 1 is consistently employed in both model adaptation and prompting method adaptation for all problems. Please let us know if we misunderstood your question.
>
> ---
>
> **Q1: Results with a general AS solver list and a more comprehensive comparison with baselines.**
>
> Comprehensive comparison with baselines is shown in Table 1. Since there are three types of adaptation designed based on the Adaptive-Solver framework, we did conduct numerous ablation experiments to verify their effectiveness respectively and analyze their functionality.
> The results with a general AS solver list and a more comprehensive comparison with baselines, are already presented in Table 1. Please let us know if we did not fully answer your question.
>
> **Q2: Table 2 only uses ZeroCoT for prompting and only the model adaptation is explained.**
>
> We explain why Table 2 only uses ZeroCoT for prompting and only the model adaptation is explained.
> In this paper, the objectives of model adaptation and two other types of adaptation are distinct: the former aims to reduce costs, while the latter aims to enhance performance. As a result, we investigate them separately. Table 1 presents the experimental results of prompting method adaptation and decomposition granularity adaptation.
> Table 2 presents the experimental results of model adaptation, focusing on the analysis of model adaptation.
>
> In the context of model adaptation, our goal is to explore whether combining a weaker LLM and a stronger LLM can significantly reduce costs while maintaining performance. Therefore, the baselines are the methods using only the weaker LLM and using only the stronger LLM. Since the variable here is the LLM model, we apply a consistent prompt (i.e., ZeroCoT) to all methods. Other prompting methods, such as PHP and SC, incur costs several times that of ZeroCoT, and factors affecting their inference costs are multifaceted, making them unsuitable for comparing the inference costs of models.

---

> > ### Comment · Reviewer_uCwU · 2023-11-23
> >
> > I would like to thank the authors for their responses.
> >
> > I did misunderstand Table 2 and thank you for the clarification. It would be beneficial to use AS-PD throughout when indicating such.
> >
> > After reading through the responses and the other reviewer comments, I feel that the paper is interesting but might not be ready for publication. For me in particular, I think that the empirical results are not significant enough in cases even though i acknowledge that the cost savings are significant. The results are collated in a way that makes it hard to understand and distill the key takeaways.
> >
> > That being said, I will have no problem if this paper were to be accepted. The cost savings are significant and the ideas presented are reasonable. I encourage the authors to fix some of the writing and resubmit in case this were to not be accepted.

---

> > > ### Author Response · Authors · 2023-11-23
> > >
> > > Dear Reviewer uCwU,
> > >
> > > We greatly appreciate your time and effort in reviewing our paper. We will improve the paper further in accordance with your suggestions.

---

### Official Review · Reviewer_xp3G · 2023-11-08

**Soundness:** 3 good
**Presentation:** 3 good
**Contribution:** 2 fair
**Rating:** 6
**Confidence:** 4

**Summary:**

This paper presents and demonstrates a simple algorithm for achieving a better cost-accuracy trade-off for reasoning tasks with LLMs. The high-level idea is to construct a cascade using different models, prompts, and/or granularities of decomposition. A crucial element is the ability to evaluate whether a solution is likely to be correct, which is achieved using the consistency of multiple samples at some non-zero temperature. The method is evaluated on a collection of reasoning datasets, and is shown to achieve a significant reduction in cost, sometimes even achieving an increase in accuracy.

**Strengths:**

1. Baselines and components are all highly recent (2022-2023).
1. Considers three different types of solver adaptation. Judicious choice of solvers in experiments.
1. Significant reductions in cost achieved. For those employing LLMs, it is useful to be aware of the efficacy of a cascade using the consistency check.
1. Ablative studies are well designed and well presented.

**Weaknesses:**

1. Little technical novelty - mostly an empirical study.
1. It seems like the temperature could have a significant impact on the consistency check, however there was no study of the effect of varying temperature.
1. It would be useful to present the ROC curve (FPR-FNR tradeoff) of the consistency check for each solver, ideally at a range of temperatures.
1. While Figure 3b shows which of the 3 decomposition prompts was used, it would be useful to know how many solvers were tried for each experiment, and how often the cascade "dropped through" to the final solver.
1. It would be good to include a discussion of determining an optimal cascade (perhaps assuming that the errors of different models are independent) per budget for a given dataset.

Suggestions (no need to address):
1. For a scientific context, I would tone down some of the grandiose language ("this innovative method represents a significant step in dynamic strategy selection", "holding vast implications for the realm of artificial intelligence").
1. Personally, I don't like the use of the word "solver" for the current purpose. Possible alternatives: tactic, strategy, protocol.

**Questions:**

Mainly just address weaknesses listed above. Additional questions:

1. Which model does each method use in Table 1? Could include this in the caption.
1. It's unfortunate that OpenAI's pricing affects the "cost saving"; changes in pricing will change the results. Is there any way around this? Is it possible to obtain flops (or kWh, but that too is technology dependent)? Otherwise at least note that this uses pricing as at [date].

---

> ### Author Response · Authors · 2023-11-21
> **Response to Reviewer xp3G (1/3)**
>
> We appreciate your detailed review and constructive feedback. We respond to your questions as follows:
>
> **W1: Little technical novelty - mostly an empirical study.**
>
> The technical novelties of our work are summarized as follows:
>
> (1) **General and flexible framework**. We propose a novel and general Adaptive-Solver framework. It is adept at strategically selecting the optimal solving methodologies tailored to the intrinsic characteristics of a given problem. This framework is flexible to support different types of adaptation (i.e., LLM model, prompting method, decomposition granularity).
>
> (2) **Cascade of LLMs reduces the cost**. The high cost is a persistent challenge when applying LLM in real-world scenarios. Our method offers a significant reduction (up to 50%) in cost while preserving performance, by cascading weaker models with stronger models. This approach represents a novel method among existing efforts aimed at enhancing the efficiency of LLMs.
>
> (3) **Adaptation of decomposition granularity**. Decomposition granularity adaptation represents a novel concept that was not previously explored in the Least-to-Most (L2M) method. Although decomposition granularity affects the performance apparently, adapting decomposition granularity tailored to each problem is a non-trivial task. In this paper, we introduce a straightforward yet effective method. It involves crafting different L2M's variants to guide an LLM in decomposing a problem at different levels of granularity.
>
> **W2: No study of the effect of varying temperature.**
>
> According to your constructive suggestion, we study the effect of varying temperature. We examine how the performances of the three types of adaptation vary with the sampling temperature. The results are presented in **Table 11 (Appendix A.5)** of the revised paper and also shown as below.
>
> Table 11: Effect of varying temperature on performance (i.e., Accuracy) of different adaptations on the GSM8K dataset. AS-P uses the solver list [COT*, L2M*], AS-D uses the solver list [(L2M*, $d_1$), (L2M*, $d_2$), (L2M*, $d_3$)], AS-M uses the solver list [GPT3.5*, GPT4]. T: Temperture.
>
> | Method | T=0.4 | T=0.7 | T=1.0 |
> | :---: | :--: | :--: | :--: |
> AS-P | 84.2 | 85.7 | 85.7 |
> AS-D | 85.9 | 87.5 | 87.2 |
> AS-M | 92.5 | 93.5 | 95.5 |
>
> Across all three adaptation types, as the temperature increases the performances consistently get higher. This phenomenon can be attributed to the fact that a higher sampling temperature enhances answer diversity. This potentially improves the precision of consistency checks in classification and gives more chances to answer a question.
> We have added this experiment into Appendix A.5 of the revised version of this paper.
>
> **W3: ROC curve of the consistency check at a range of temperatures**
>
> According to your insightful suggestion, we present the ROC curve of the consistency check (sample size = 3) for four different prompting methods at a range of temperatures {0.4, 0.7, 1.0}. The experiment was conducted with the GPT-3.5 model, on the GSM8K dataset. As shown in **Figure 4 (Appendix A.6)** of the revised paper, as the temperature increases, the performances (i.e., AUC) of the consistency check also improve but converge when temperature (at 0.7) gets high enough.
> We have added this experiment into Appendix A.6 of the revised version of this paper.

---

> > ### Author Response · Authors · 2023-11-21
> > **Response to Reviewer xp3G (2/3)**
> >
> > **W4: About Figure 3b, how many solvers were tried for each experiment, and how often the cascade "dropped through" to the final solver**
> >
> > The required information may not be critical to the experiment where Figure 3b locates at. We would like to first clarify the intention behind creating Figure 3b and then provide the required information.
> >
> > (1) **The intention behind creating Figure 3b**. Figure 3b is primarily used to analyze the reasons behind the effectiveness of the decomposition granularity adaptation. Figure 3a indicates that as the question difficulty increases, the prompts with finer granularity perform better. In Figure 3b, we show how usage ratios of the prompts with different granularities vary with question difficulty. Figure 3b indicates that as the question difficulty increases, the proportion of using prompts with finer granularity also increases. Therefore, combining Figure 3a and Figure 3b, it can be understood that the reason the decomposition granularity adaptation works is that it can increase the use of more accurate prompting methods as the question difficulty increases.
> >
> > (2) **The required information may be already included**. It appears that the inquiries are more pertinent to the cost efficiency of decomposition granularity adaptation. If the question "how many solvers were tried for each experiment" refers to the average number of solvers attempted for each problem, the corresponding results are detailed in Table 8 in the Appendix A.4. If the query "how often the cascade 'dropped through' to the final solver" refers to the ratio of the final solver ultimately being employed, the value of L3 in Figure 3b actually represents this information. Please let us know if we misunderstood your question.
> >
> > **W5: Discussion of determining an optimal cascade per budget for a given dataset**
> >
> > We provide two approaches to determine an optimal cascade per budget for a given dataset.
> >
> > (1) The first method is a broad approach applicable for all the types of adaptation. We can sample a small subset of the dataset as a validation set and try different permutations and combinations of solvers to decide which solvers to include and the optimal order in the solver list that meet the budget requirements.
> >
> > (2) The second method specifically targets model adaptation, with the goal of maintaining performance while minimizing costs. Typically, as the model's capabilities increase, so does the associated cost. Hence, a pragmatic strategy involves constructing the solver list by arranging them from weaker to stronger based on the model's ability.
> >
> > We have added this discussion into Appendix A.7 of the revised version of this paper.

---

> ### Author Response · Authors · 2023-11-21
> **Response to Reviewer xp3G (3/3)**
>
> **Q1: Which model does each method use in Table 1? Could include this in the caption.**
>
> In Table 1, we used GPT-3.5-turbo-0301 as LLM model for all prompting methods, as indicated in Appendix A.2 "implementation details" of the paper.
> We have included the model information in the caption of Table 1.
>
> **Q2: OpenAI's pricing affects the "cost saving"**
>
> Thanks for the suggestion. As OpenAI's pricing might change, below, we present cost analysis independent of pricing.
>
> (1) **Cost analysis with varying API cost**. The change in OpenAI's pricing does impact the specific value of cost savings. Utilizing the FLOPs of LLM during the inference phase as a metric to measure cost savings is a reasonable and practical method. The FLOPs per token of GPT3.5 and GPT4 are mysterious to public, but it should be positively related to API Cost. Considering the variation of API Cost over time, we can represent it with a variable and find the conditions under which our method can save costs.
>
> We documented the average number of input tokens and output tokens per problem of GPT4 and our method [GPT3.5*, GPT4], as presented in the following table.
>
> | | | GSM8K-200 | SVAMP-200 | AQuA |
> | :---:| :---:| :---:| :---:| :---:|
> GPT 4 |	# input token of GPT 4 | 300.3 | 209.5 | 252.1 |
> | | # output token of GPT 4 |	154.5 | 104.2 | 217.4 |
> [GPT3.5*, GPT4] | # input token of GPT3.5* / GPT4 | 897.7 / 102.1 | 609.9 / 69.3 | 532.6 / 139.2 |
> | | # output token of GPT3.5* / GPT4 | 557.5 / 56.6 | 359.7 / 34.4 | 982.9 / 127.4 |
>
>
> Let the API cost per token of GPT3.5 and GPT4 be S and L (Suppose the costs for input token and output token are the same, for simplicity). Take GSM8K dataset as example, we can calculate the overall API cost of GPT4 and [GPT3.5*, GPT4] as follows:
>
> API cost of GPT 4 = L * (300.3 + 154.5) = 454.8 * L
>
> API cost of [GPT3.5*, GPT4] = S * (897.7 + 557.5) + L * (102.1 + 56.6) = 1455.2 * S +158.7 * L
>
> Relative Cost Saving = [454.8 * L - (1455.2 * S +158.7 * L)] / 454.8 * L = 0.65 - 3.2 * (S/L)
>
> Let 0.65 - 3.2 * (S/L) > 0, get L/S > 4.93.
>
> This indicates that, as long as the API cost per token of GPT4 is more than 4.93 times that of GPT3.5, our method can save costs.
>
> (2) **Specification of the used API cost**. We have specified the experiment date and the corresponding API prices at that time in Appendix A.2 "implementation details".
>
> ---
> **S1: Some of the language is grandiose**
>
> A: Thank you for your suggestion. We have modified the language in the revised version of this paper.
>
> **S2: The use of the word "solver" and possible alternatives: tactic, strategy, protocol.**
>
> A: Thank you for your suggestion. In our view, "tactic, strategy, protocol" bears a closer resemblance to the concept of the "prompting method" in this paper. However, within the given context, the term "solver" carries a broader meaning. A solver encompasses all elements integral to problem-solving, including the LLM model, prompting techniques, decomposition strategies, and so forth.

---

> > ### Author Response · Authors · 2023-11-23
> >
> > Dear Reviewer xp3G,
> >
> > We sincerely appreciate your insightful and constructive feedback. We have carefully considered your comments and have tried to comprehensively address your questions in our response. As the discussion period approaches its conclusion, we kindly inquire if there are any additional questions or if our responses have sufficiently addressed your concerns. We would be happy to discuss any additional feedback you may have.